# Weatherproofing Retrieval for Localization with Generative AI & Geometric Consistency

**Yannis Kalantidis**[*]   **Mert Bülent Sarıyıldız**[*]   **Rafael S. Rezende**

**Philippe Weinzaepfel**   **Diane Larlus**   **Gabriela Csurka**

NAVER LABS Europe

## Abstract

State-of-the-art visual localization approaches generally rely on a first image retrieval step whose role is crucial. Yet, retrieval often struggles when facing varying conditions, due to *e.g.* weather or time of day, with dramatic consequences on the visual localization accuracy. In this paper, we improve this retrieval step and tailor it to the final localization task. Among the several changes we advocate for, we propose to synthesize variants of the training set images, obtained from generative text-to-image models, in order to automatically expand the training set towards a number of nameable variations that particularly hurt visual localization. These changes result in **Ret4Loc**, a training approach that learns from such synthetic variants together with real images and that leverages geometric consistency for filtering and sampling. Experiments show that it leads to large improvements on multiple challenging visual localization and place recognition benchmarks. Project page: https://europe.naverlabs.com/ret4loc

## 1 Introduction

Visual localization, the task of estimating the camera pose for a novel view of a known scene, is a core component of the perception system of autonomous platforms. It generally relies on image retrieval techniques that provide an approximate pose estimate that is further refined. As Humenberger et al. (2022) show, this retrieval step significantly impacts the overall visual localization accuracy, yet an improved retrieval performance does not necessarily imply a better visual localization. This is in part explained by the second stage of visual localization; The pose estimated by the retrieval step is refined *e.g.* using Structure-from-Motion, so it can be, to some extent, resilient to isolated errors. Yet, when a challenging visual query leads to retrieved candidates that are wrong in a consistent manner, no recovery is possible. This is why this paper does not study retrieval in isolation, but through the prism of results from the full localization pipeline.

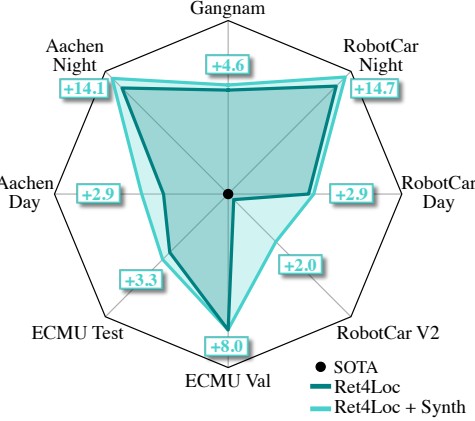

Figure 1: **Gains in localization accuracy** using our Ret4Loc models compared to the state of the art (**black dot**). We show results for our best models trained on *only real* (**Ret4Loc**) or *real and synthetic images* (**Ret4Loc + Synth**), for 7 outdoor and 1 indoor dataset splits. Axes in log-scale.

One of the major challenges faced by retrieval models is linked to appearance variations caused by lighting, weather and seasonal changes. This has been known to the community for a while, as shown by the many methods tailored to one of those variations (see Sec. 5), and is typically referred to as *long-term visual localization* (Sattler et al., 2018; von Stumberg et al., 2020; Toft et al., 2022).

---

[*]Equal contribution

In this paper, instead of building a custom retrieval model for each single such variation, we propose to address all of them jointly. We start from state-of-the-art landmark retrieval and place recognition method HOW (Tolias et al., 2013) and design retrieval models tailored to the task of visual localization. We will refer to those as Retrieval for Visual Localization or ***Ret4Loc*** models. First, we improve the training process and use strong data augmentations, a change proved crucial to the final localization performance. Second, we go beyond generic pixel transformations, and explicitly tackle the domain shifts relevant to the task: the ones related to weather, season, and time of day.

To that end, we leverage the fact that challenging conditions for visual localization can be expressed in *plain text*. Thanks to impressive progress in generative modeling, it is now possible to alter images in a realistic manner using a textual prompt (Brooks et al., 2023). Here, we directly use the adversarial conditions identified by the community (Toft et al., 2022) as a set of textual prompts. They correspond to changing illumination (day to dusk/dawn/night), weather (clear to sunny/rainy/snowy), and season (summer to winter). We use those to generate synthetic variants that extend the set of images used to train retrieval models, improving their resilience to such changing conditions.

The resulting extended training set is indeed more diverse, and can be used to train models more robust to such conditions. Yet, off-the-shelf generative models provide no guarantee that these high-level image manipulations will preserve location-specific information, crucial for visual localization: they might alter the content on top of applying the requested domain shift. To mitigate this issue, we leverage the fact that the training relies on image pairs and propose several geometry-aware strategies to filter the extended dataset and even to alter the training process itself.

Ret4Loc is evaluated following two protocols: a faster one that estimates the query's pose using the top retrieved images, and a slower but more accurate one based on Structure-from-Motion that uses the 3D map of the scene. We show that our retrieval models achieve consistent gains in both cases, for many standard datasets. Fig. 1 summarizes our gains under the first protocol.

**Contributions.** We tailor the training of retrieval models to the challenging task of long-term visual localization. We use language-based data augmentation to improve the robustness of retrieval models to adversarial daytime, weather and season variations. We further introduce geometry-based strategies for filtering and sampling from the pool of synthetic images and report significant gains over the state of the art on six common benchmarks. To our knowledge, this is the first time language-based generative models are used for generating and validating targeted scene alterations while preserving the crucial visual characteristics needed for an instance-level recognition task.

## 2 RET4LOC: LEARNING RETRIEVAL MODELS FOR VISUAL LOCALIZATION

We start from the state of the art in landmark retrieval and gradually tailor it to visual localization. We observe that the training of top models is lacking, *i.e.* data augmentation and a more adaptive learning framework are missing. On top of standard pixel-level augmentations, we explore more drastic synthetic variations specifically created to target the domain shifts long identified by the visual localization community. Next, because the extreme nature of those variations conflicts with the level of precision required in retrieval for localization, we describe a geometric consistency score that can be used for selecting and sampling the synthetic data.

### 2.1 TAILORING EXISTING RETRIEVAL METHODS TO VISUAL LOCALIZATION

After benchmarking a broad range of retrieval methods for visual localization, Humenberger et al. (2022) exposed the critical role of the retrieval step on the final localization performance. We extended their study and found that more recent landmark retrieval methods like HOW (Tolias et al., 2020) and FIRe (Weinzaepfel et al., 2022) are even more competitive (see Sec. 3.1). We therefore build our retrieval framework on top of HOW which uses a contrastive loss over a training set of matching pairs depicting the same landmark. The loss is applied on aggregated global features at training time. At test time, HOW is typically paired with ASMK (Tolias et al., 2013) matching.

Let us assume access to a training set $\mathcal{D}$ of images suitable for landmark retrieval with matching pairs, *i.e.* images depicting the same landmark or scene. Formally, the training set can be seen as a set of training tuples $\mathcal{U}_{qp}$, each composed of a matching pair $(q, p)$, built from a *query* image and a *positive* image for that query, and a small set of $M$ *negatives* images, *i.e.* $\mathcal{U}_{qp} = (q, p, n_1, ..., n_M)$.

Let $f(x)$ be the aggregated feature vector for image $x$. It is computed as a weighted average of all local features, the weights being proportional to the local features' $\ell_2$ norm, and then $\ell_2$-normalized. Given a matching pair $(q, p)$ and the corresponding tuple $\mathcal{U}_{qp}$, the loss used for training by Tolias et al. (2020) is given by:

$$\mathcal{L}_c(\mathcal{U}_{qp}) = \|f(q) - f(p)\|_2^2 + \sum_{m=1}^{M} \left[ \mu - \|f(q) - f(n_m)\|_2^2 \right]^+, \qquad (1)$$

where $\mu$ is a margin hyper-parameter, and $[\cdot]^+$ denotes the positive part function $max(0, \cdot)$.

**Towards a better retrieval step for visual localization.** Landmark retrieval methods tend to only use basic data augmentations (Berton et al., 2022), while state-of-the-art methods such as HOW (Tolias et al., 2013) do not even use any during training. We tailor HOW's training process to long-term visual localization by learning more robust features. First, we apply domain randomization, that has been shown by Volpi et al. (2021) to improve resilience to domain shifts. For this, we apply a random cropping mechanism and we use AugMix (Hendrycks et al., 2020) that mixes multiple augmentations. Other improvements include using the AdamW (Loshchilov & Hutter, 2019) optimizer and training for longer with a cosine learning rate schedule.[1]

We refer to models trained with this setup as Retrieval for Visual Localization or Ret4Loc. In this paper we explore Ret4Loc models that follow HOW for training (**Ret4Loc-HOW**), but other models, *e.g.* FIRe, could have been used instead. Ret4Loc-HOW models boost visual localization performance for both indoor and outdoor benchmarks and hence greatly improve over the state of the art for any type of localization (see results in Sec. 3.1).

Although effective, augmentations described above are not tailored to visual localization. Next, we leverage the fact that the main challenges faced by long-term visual localization can be *named*.

## 2.2 Synthesizing variants for diverse weather & time conditions

Our goal is to improve the resilience of retrieval models to challenging conditions such as changes related to seasons, weather, and time of day. Those changes are not only predictable, they can also be clearly and concisely expressed via natural language. We therefore propose to use generative models to synthesize such challenging scenarios. DALL-E (Ramesh et al., 2021) or Stable Diffusion (Rombach et al., 2022) have demonstrated impressive text-to-image generation ability. Methods like InstructPix2Pix (Brooks et al., 2023) extend them to *altering* images via a textual prompt.

Given a set of textual prompts for scenarios we care about, we employ InstructPix2Pix to generate multiple synthetic *variants* for every image in our training set. Formally, let $g(\cdot)$ be a generative model that takes as input an image $x$ and a textual prompt $t$, and produces $\tilde{x}_t$, a synthetic variant of image $x$ with respect to textual prompt $t$, *i.e.* $\tilde{x}_t = g(x; t)$. Given our training set $\mathcal{D}$ and a set of $T$ textual prompts $\mathcal{T} = \{t_1, .., t_T\}$, we generate an extended dataset which contains the original images as well as their $T$ variants $\tilde{x}_t = g(x; t)$, for every image in $\mathcal{D}$ and for all $t \in \{1, .., T\}$.

The seminal Robotcar Seasons visual localization benchmark (Maddern et al., 2017) has identified the most common domain shifts for long-term visual localization. We use those to define a set of 11 textual prompts to create synthetic variants for our training set: `at dawn`, `at dusk`, `at noon`, `at sunset`, `in winter`, `in summer`, `with rain`, `with snow`, `with sun`, `at night with rain`, `at night`. Fig. 2a shows a few generated variants for 3 images from the training set. The complete set of 11 variants can be found in Appendix C.

## 2.3 Training with synthetic variants

Once we have generated multiple variants for all training images, we sample them to form *synthetic tuples* that we use during training together with the original ones. To obtain a synthetic tuple, we propose to substitute the query image $q$ and all negatives of an original training set tuple with their synthetic variants. Note that the positive image $p$ is never substituted and therefore the matching pair of each synthetic tuple contains both an altered and an original image. To make sure negatives in the tuple remain challenging, we choose variants consistently across a tuple, *i.e.* we use variants from the same textual prompt for the query and the negatives.

---

[1] Further implementation details and hyper-parameters of our training setup can be found in Appendix A.

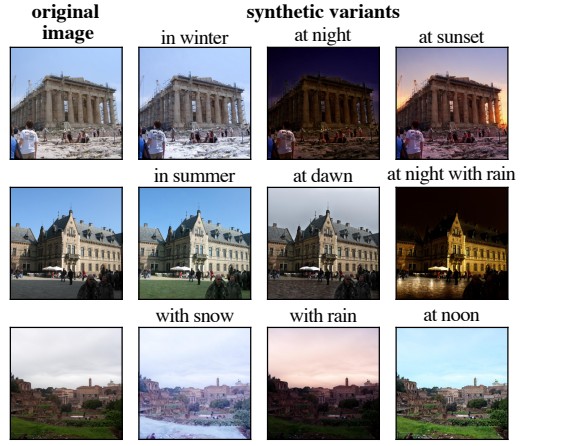

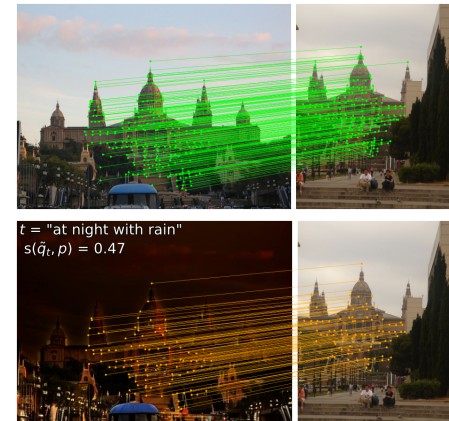

(a) Synthetic variants for the images shown on the left.

(b) Geometric correspondences computed before (top) and after (bottom) synthetic alteration.

Figure 2: (Left) Synthetic variants for several prompts (the full set of variants is shown in Fig. 7). (Right) Estimated local correspondences between two matching images before and after alteration.

Formally, given a tuple $\mathcal{U}_{qp}$ corresponding to the matching pair $(q, p)$, we produce a synthetic tuple $\tilde{\mathcal{U}}_{qp}^t = (\tilde{q}^t, p, \tilde{n}_1^t, ..., \tilde{n}_M^t)$ for the variant corresponding to textual prompt $t$ by replacing $q$ with $\tilde{q}_t$ and all the negatives $n_m$ with $\tilde{n}_m^t$. Intuitively, this means that, when a synthetic tuple is used in the contrastive loss of HOW (Eq. (1)), the first part of the loss, which uses a synthetic $\tilde{q}^t$ with an unaltered $p$, brings the representations of the original positive image $p$ and of the synthetic variant of the query $\tilde{q}^t$ close to each other. At the same time, the loss pushes the synthetic variant of the query $\tilde{q}^t$ away from the representations of all the synthetic negative images $\tilde{n}_m^t$, in the domain corresponding to the selected textual prompt. This aligns with our intuition that i) the query feature should be invariant to the different domain shifts described by the prompts and simultaneously ii) given any domain shift, the query feature should be different enough from its associated negatives.

We experimented with several ways of using synthetic tuples during training. The most effective one uses both the original tuple and one or more synthetic tuples sampled from the set of possible ones for the different prompts. By default, we uniformly sample $K$ textual prompts to select synthetic tuples among $T$ options (we discuss a geometry-based sampling alternative next in Sec. 2.4). Formally, let us consider a set of $K$ synthetic tuples $\{\tilde{\mathcal{U}}_{qp}^1, ..., \tilde{\mathcal{U}}_{qp}^K\}$ corresponding to $K \geq 1$ selected variants. We can define the set of tuples $\tilde{\mathcal{U}} = \{\mathcal{U}_{qp}, \tilde{\mathcal{U}}_{qp}^1, ..., \tilde{\mathcal{U}}_{qp}^K\}$ that contains the original tuple $\mathcal{U}_{qp}$ and all synthetic ones. The loss is then computed over this extended set of tuples.

**Loss computation with synthetic tuples (Ret4Loc-HOW-Synth).** A simple way to compute the overall loss for the extended tuple set $\tilde{\mathcal{U}}$ would be to sum the individual losses incurred from the real and synthetic tuples, *i.e.* $\mathcal{L}_c(\tilde{\mathcal{U}}) = \mathcal{L}_c(\mathcal{U}_{qp}) + \sum_k \mathcal{L}_c(\tilde{\mathcal{U}}_{qp}^k)$. An alternative that performs better in practice is to first aggregate features from all variants of each image in the original tuple independently, and then compute a single loss on the aggregated features. More precisely, let $Q, P, N_m$ respectively be sets of features extracted for the corresponding query, positive[2] and each of the negative images that appear in the extended tuple set $\tilde{\mathcal{U}}$. Let $\phi$ be an aggregation function, *e.g.* simple averaging. The function $\phi$ produces a single aggregated feature vector for each set of features $Q, P$, and $N_m$. The loss then can be applied on the aggregated vectors, *i.e.*:

$$\mathcal{L}_c(\tilde{\mathcal{U}}) = \|\phi(Q) - \phi(P)\|_2^2 + \sum_{m=1}^{M} \left[ \mu - \|\phi(Q) - \phi(N_m)\|_2^2 \right]^+. \tag{2}$$

In this loss, all original and synthetic query, positive and negative images impact each other, *i.e.* tuples are not treated independently anymore. By default we use the loss on the aggregated features, *i.e.* the loss from Eq. (2), for Ret4Loc models. We ablate this choice in the experiments.

---

[2]Although we do not substitute $p$ with synthetic variants, we do sample different augmentations of the positive for every synthetic tuple and still end up with a set of features in P that we can aggregate.

## 2.4 Incorporating geometric constraints for synthetic variants

In this section we describe an automatic way of assessing to which extent important characteristics of the scene are retained during the image generation process, and propose two strategies that incorporate this information during training. We argue that the geometric consistency of a matching pair should not change significantly when the query image is replaced by a synthetic variant. This means that geometric correspondences estimated between a pair should mostly remain after altering one of the images with a generative model. Using this insight, we define a *geometric consistency score* to assess the degree to which a synthetic pair preserves location characteristics shared across the matching images.

Let $(q, p)$ be a matching pair from the training set, and $(\tilde{q}^t, p)$ its corresponding synthetic pair where the query is replaced by a synthetic variant obtained with textual prompt $t$. Let $c(\cdot, \cdot)$ denote a matching function that returns a set of *correspondences* $C_{qp}$, *i.e.* a set of geometrically consistent local matches between $q$ and $p$. We can also compute correspondences $C_{\tilde{q}^t p}$ for the synthetic pair and then filter them such that only correspondences that existed in the original pair $(q, p)$ are kept: $||C_{\tilde{q}^t p}||' = ||C_{\tilde{q}^t p}|| \cap ||C_{qp}||$. We propose to use the ratio of correspondences remaining after replacing $q$ by $\tilde{q}^t$ in the tuple, $s(\tilde{q}^t, p) = \frac{||C_{\tilde{q}^t p}||'}{||C_{qp}||}$, as a geometric consistency score. This score ranks synthetic pairs according to the level of preservation of their geometry. In Fig. 2b we show correspondences between images of a matching pair, before and after replacing the query with one synthetic variant. Next, we present two ways of exploiting this score during training.

**Synthetic tuple filtering (Ret4Loc-HOW-Synth+).** We can directly use the geometric consistency score above to select the synthetic tuples that are used during training. The score is in $[0, 1]$ and essentially measures the percentage of correspondences remaining for the matching pair after the query is altered. One can set a threshold $\tau$ under which a synthetic tuple will be discarded; we otherwise refer to it as *valid*. Increasing the threshold means that we are being more and more selective. We can therefore restrict the sampling of tuples during training to only pick among the valid ones, *i.e.* when sampling synthetic tuples for an original training tuple.

**Geometry-aware sampling (Ret4Loc-HOW-Synth++).** When choosing synthetic tuples, by default, we sample a textual prompt $t \in \mathcal{T}$ uniformly. However, we now have a filtered set of tuples, each with a geometric consistency score $s$ which correlates with the level of local appearance preservation. We propose to use this score to also compute the probability of sampling variants. Sampling a valid synthetic tuple $\tilde{\mathcal{U}}_{qp}^t$ proportionally to $1/s(\tilde{q}^t, p)$ means that tuples with a larger drop in correspondences are picked more often; This weighing scheme favors valid tuples that are "harder". We refer to this as *geometry-aware sampling*. Note that this only works well in conjunction with tuple filtering, as variants with very low geometric consistency should never be considered.

## 3 Experiments

In this section, we evaluate our Ret4Loc models on visual localization and place recognition datasets. Two of those are indoor datasets while the others are outdoor ones. Although part of our contributions explicitly targets outdoor localization, we show that Ret4Loc also leads to state-of-the-art results for indoor localization. We present extended results for place recognition in Appendix B.4.

**Datasets.** Following HOW (Tolias et al., 2020), we use the SfM-120K dataset (Radenović et al., 2019) to train all Ret4Loc models. We augment SfM-120k by generating 11 synthetic variants for each image with the process described in Sec. 2.2 and use them to build synthetic tuples that are used during training. We evaluate localization accuracy on five datasets from the Visual Localization Benchmark,[3] *i.e.* RobotCar Seasons (Maddern et al., 2017), Aachen Day-Night v1.1 (Zhang et al., 2021), Extended CMU Seasons (Toft et al., 2022), Gangnam Station B2 (Lee et al., 2021) and Baidu Mall (Sun et al., 2017). For completeness, we also evaluate place recognition on the popular Tokyo 24/7 dataset (Torii et al., 2015a).

**Synthetic variant generation and geometric consistency.** We use the public InstructPix2Pix model from Brooks et al. (2023) for generating synthetic variants.[4] We use 20 inference sampling

---

[3] https://www.visuallocalization.net/
[4] https://github.com/timothybrooks/instruct-pix2pix

| | Model | VMix | Filt. | GaS | RobotCar-v2 EWB (top-1) | | ECMU-val EWB (max) | | RobotCar-Day SfM, (max) | | RobotCar-Night SfM, (max) | | Tokyo 24/7 Recall@$k$ | |
|---|---|---|---|---|---|---|---|---|---|---|---|---|---|---|
| | | | | | .5m/5° | 5m/10° | .5m/5° | 5m/10° | .25m/2° | .5m/5° | .25m/2° | .5m/5° | R@1 | R@10 |
| 1. | HOW | – | – | – | 29.8 | 74.4 | 25.9 | 81.1 | 52.6 | 81.1 | 17.4 | 35.0 | 89.2 | 96.5 |
| 2. | Ret4Loc-HOW | – | – | – | 30.6 | 75.0 | 33.4 | 88.6 | 52.8 | **81.5** | 21.8 | 40.4 | 89.2 | **97.1** |
| 3. | | – | – | – | 31.5 | 75.9 | 33.4 | 88.0 | 52.8 | 81.1 | **25.1** | **47.1** | 88.9 | 96.2 |
| 4. | | ✔ | – | – | 31.7 | 76.3 | 34.3 | 88.6 | 52.7 | 81.2 | 23.0 | 43.6 | 90.5 | 96.2 |
| | Ret4Loc-HOW + synthetic | *Variations using geometric consistency* | | | | | | | | | | | | |
| 5. | | ✔ | ✔ | – | **31.8** | **77.1** | 34.4 | 88.8 | 52.8 | **81.5** | 22.1 | 43.7 | 90.2 | 96.2 |
| 6. | | ✔ | ✔ | ✔ | 31.3 | 76.4 | **34.5** | **89.1** | **53.0** | 81.4 | 21.0 | 41.7 | **91.1** | **97.1** |

Table 1: **Impact of synthetic data and geometric consistency**. We report results for different flavors of Ret4Loc training with and without the use of synthetic variants and geometric consistency. *VMix* refers to the use of variant mixing with Eq. (2) instead of summing the different losses. *Filt.* refers to synthetic tuple filtering, *GaS* to geometry-aware sampling. Columns denoted as *max* report the top performance achieved across all different values of top-$k$, *i.e.* the best across $k \in \{1, \ldots, 50\}$.

steps for diffusion, and set the text prompt and image guidance scale to 10 and 1.6, respectively. We ran synthetic image generation as a pre-processing step, for all prompts in parallel. This took approximately a day (more details in Appendix B.6). We employ LightGlue (Lindenberger et al., 2023) as our matching function $c$ to estimate the geometric consistency score $s(\cdot, \cdot)$ that is used for filtering or sampling synthetic variants.

**Training details.** We build on the HOW codebase[5] and, unless otherwise stated, we follow the training protocol from HOW with the modifications presented in Sec. 2.1. We train ResNet50 models (He et al., 2016). We use randomly resized crops of size $768 \times 512$ during training as additional data-augmentation as well as to provide the network with fixed sized inputs. After preliminary experiments, we found that using the original plus two synthetic variants ($K$=2) perform best. Our model trains in less than 8 hours on a single A100 GPU (more details in Appendix B.6).

**Evaluation protocols.** At test-time, we observe a large variance in performance when running the exact same training setup with different seeds. To get more robust results, we report results after averaging the weights of 3 models for each setup, *i.e.* we average the weights of 3 models trained from 3 different seeds (Wortsman et al., 2022). We follow HOW and use multi-scale queries together with ASMK matching. To measure visual localization performance, we use the Kapture localization toolbox.[6] In the retrieval step of this toolbox, a retrieval model produces a shortlist of the top-$k$ nearest database images per query. For our experiments, we replace the retrieval model with the public HOW or FIRe models, or one of the different Ret4Loc flavors we designed. The rest of the pipeline, *i.e.* the pose estimation step, is shared across all experiments, and follows two of the protocols presented by Humenberger et al. (2022): a) pose approximation with equal weighted barycenter (*EWB*) using the poses of the top-$k$ retrieved database images; or b) pose estimation with a global 3D-map (*SfM*) and R2D2 (Revaud et al., 2019b) local features. More implementation and protocol details are presented in Appendix A.

## 3.1 RESULTS

In Tab. 1 we present results for several Ret4Loc-HOW models trained under different setups. We see that the basic Ret4Loc-HOW setup (row 2) already brings some consistent gains over HOW. Using synthetic variants during training further improves performance even without any filtering (rows 3-4). We also see that mixing the variants generally improves performance (rows 3 *vs*. 4) in all datasets apart from RobotCar-Night. Inspecting our complete set of results on 6 datasets, overall we see consistent gains when incorporating synthetic data during training. From rows 5 and 6, it appears that we can get the top performance in all datasets except RobotCar-Night by incorporating geometric information, either as a way of filtering or also sampling the synthetic variants.

---

[5] https://github.com/gtolias/how
[6] https://github.com/naver/kapture-localization

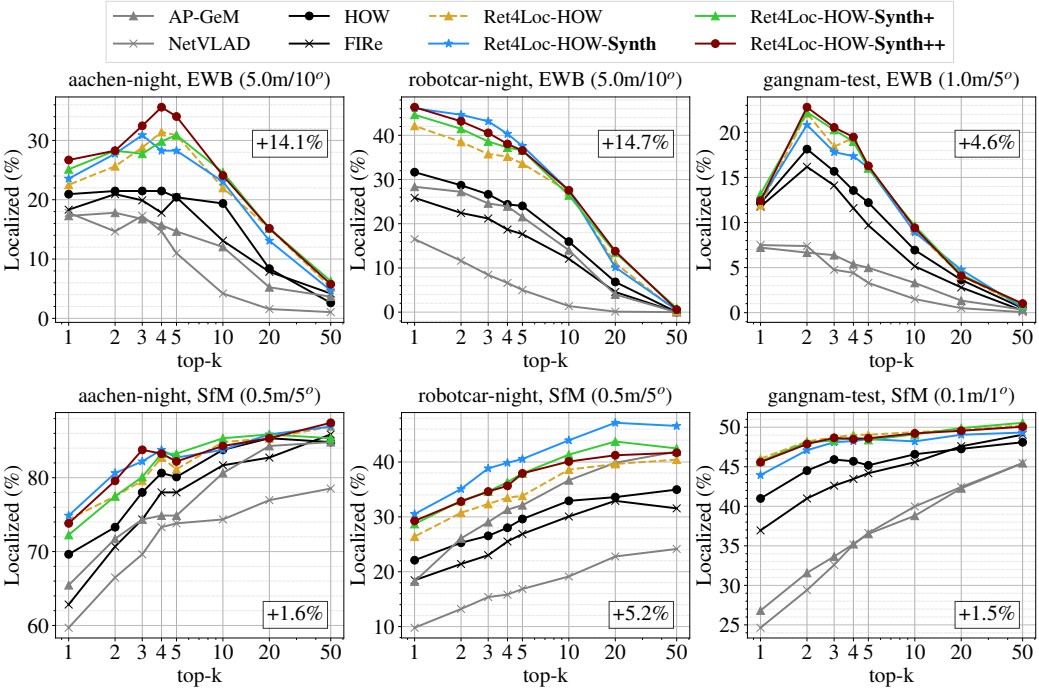

Figure 3: **Localization accuracy as a function of the top-$k$ retrieved images** for Ret4Loc models and the state of the art. *Top*: Pose approximation (EWB) protocol. *Bottom*: Structure-from-Motion (SfM) based protocol. Ret4Loc-HOW-**Synth** variations using geometric consistency are denoted with a "+". In each plot, we further denote the top gains achieved using Ret4Loc models over the best performing competing method. See Appendix B.2 for a complete set of results on more datasets.

| | | RobotCar-v2 | | ECMU-test | | Tokyo 24/7 | |
|---|---|---|---|---|---|---|---|
| Model | | EWB (top-1) | | EWB (top-1) | | Recall@$k$ | |
| | | .5m/5° | 5m/10° | .5m/5° | 5m/10° | R@1 | R@10 |
| *Single stage retrieval methods* | | | | | | | |
| 1. GCL* | | 21.9 | 74.7 | 18.2 | 74.9 | 69.5 | 85.1 |
| 2. FiRE‡ | | 26.0 | 74.0 | 24.4 | 83.6 | 84.8 | 92.7 |
| 3. HOW‡ | | 29.8 | 74.4 | 30.5 | 87.9 | 89.2 | 96.5 |
| 4. Ret4Loc-HOW ‡ | | 30.6 | 75.0 | 32.6 | 90.5 | 89.2 | **97.1** |
| 5. Ret4Loc-HOW-**Synth+** ‡ | | **31.8** | **77.1** | 34.1 | 91.0 | 90.2 | 96.2 |
| 6. Ret4Loc-HOW-**Synth++** ‡ | | 31.3 | 76.4 | **34.2** | **91.2** | **91.1** | **97.1** |
| *Retrieval methods with re-ranking* | | | | | | | |
| 7. SP-SuperGlue* | | 35.4 | 85.4 | 30.7 | 96.7 | 88.2 | 90.2 |
| 8. DELG* | | 8.4 | 76.8 | 21.1 | 93.6 | 95.9 | 97.1 |
| 9. Patch NetVLAD* | | 35.3 | **90.9** | 36.2 | 96.2 | 86.0 | 90.5 |
| 10. Ret4Loc-HOW-**Synth++** ‡ | | **38.6** | 84.2 | **42.4** | **98.3** | **97.5** | **98.4** |

Table 2: **Comparison to the state of the art.** ‡ denotes methods using ASMK. * denotes results from GCL.

| Method | Purity | ARI | NMI |
|---|---|---|---|
| HOW | 0.82 | 0.75 | 0.95 |
| Ret4Loc-HOW | 0.85 | 0.78 | 0.96 |
| Ret4Loc-HOW-Synth | 0.89 | **0.85** | **0.97** |
| Ret4Loc-HOW-Synth+ | **0.90** | **0.85** | **0.97** |
| Ret4Loc-HOW-Synth++ | 0.89 | **0.85** | **0.97** |

Table 3: **Feature space analysis**. We measure how well an image and its synthetic variants get clustered in the different Ret4Loc feature spaces. We sample 1000 images from SfM-120k, consider each image and its 11 variants a "class" and cluster their features into 1000 clusters. We report purity, adjusted random index (ARI) and normalized mutual information (NMI).

To complement Tab. 1, Fig. 3 presents localization results across multiple top-$k$ results. Here, we only compare different variants of our method to HOW and FIRe, as they already surpass the older methods evaluated by Humenberger et al. (2022). We clearly see the superiority of all Ret4Loc models across protocols and top-$k$ values, with gains exceeding $10\%$ in localization accuracy for the case of night queries. The right-most plots show results on Gangnam Station B2 (Lee et al., 2021), a dataset for *indoor* localization. We see that strong augmentations help indoor localization as well. It is also nice to observe that the addition of synthetic data tailored to outdoor domain shifts does not hurt indoor performance, making Ret4Loc *the state-of-the-art retrieval method for both indoor and outdoor localization*. Complete sets of results are presented in Appendix B.2.

**Comparison to the state of the art.** We found two methods designed for landmark image retrieval, HOW (Tolias et al., 2020) and FIRe (Weinzaepfel et al., 2022), to be state-of-the-art for the localization benchmarks we considered, a finding also confirmed by Aiger et al. (2023). We also compare Ret4Loc to recent place recognition methods like GCL (Leyva-Vallina et al., 2023). and TransVPR (Wang et al., 2022). We also report results after a complementary geometric re-ranking step, *i.e.* after re-ranking the top 100 retrieval results from our top Ret4Loc model using Light-Glue (Lindenberger et al., 2023). We compare to SP+SuperGlue (Sarlin et al., 2020), DELG (Cao et al., 2020) and Patch NetVLAD (Hausler et al., 2021). Details are provided in Appendix A.5.

In Tab. 2 we compare Ret4Loc-HOW to the most recent published results on visual localization. For this reason, we restrict the comparison to the protocols used in related works, *i.e.* EWB and top-1. We see that our method outperforms by a large margin the very recent GCL, a method also using contrastive learning. This means that, in theory, training with synthetic variants could also benefit GCL. We considered methods with reranking in a separate part of the table as they benefit from an additional reranking step, that could be applied to all single stage retrieval methods. We also clearly mark methods that use ASMK matching, that seems to help for the task. Concurrent work by Aiger et al. (2023) replaces ASMK matching with CANN and shows large gains for HOW and FIRe. We see CANN as orthogonal and complementary to Ret4Loc training.

In Fig. 1 we show a birds-eye view of the significant gains over the state of the art (SOTA) one can get with and without the use of synthetic data, *i.e.* using the Ret4Loc-HOW-**Synth++** and Ret4Loc-HOW models, respectively. We denote as SOTA the best performance per dataset across all single stage retrieval methods, *i.e.* in practice either HOW or FIRe. We report results using the EWB protocol, for the highest accuracy threshold and show the radial axes in log scale.

## 4 DISCUSSION AND ANALYSIS

**Feature space analysis.** To better understand how the representation space changes from the baseline HOW to Ret4Loc-HOW, and by using synthetic data and geometry, we performed some feature analysis on a subset of SfM-120k validation images and all their synthetic variants. We measure alignment and uniformity (Wang & Isola, 2020) for features from different retrieval models (Fig. 6 in the Appendix) and observe that the use of synthetic data leads to a better balance between these two quantities, as opposed to Ret4Loc-HOW and to the baseline HOW. We also performed a clustering analysis (Tab. 3) and found that, among other metrics, cluster purity improves by 2.5% in Ret4Loc-HOW over the baseline, and then by 3-4% more when using synthetic data and geometry. More details on this analysis can be found in Appendix B.5.

**Potential alternatives to the generation of synthetic variants.** The textual prompts we use are concise and only contain one main alteration. Following recent works on prompt engineering, our method could be extended by creating more intricate prompts (Zhou et al., 2022); we find such explorations beyond the scope of this paper. We use InstructPix2Pix, but any generative model that can alter an image using a textual prompt could be used instead. The set of prompts can be replaced or extended to include other domain shifts as long as they can be expressed in natural language.

**Synthetic variants and indoor localization.** From Sec. 3.1 we see that the synthetic variants created with outdoor localization in mind also benefit indoor localization datasets. This is consistent with findings that extreme data augmentation helps generalization (Sariyildiz et al., 2023; Volpi et al., 2021). It is worth mentioning that the SfM-120k dataset also contains images taken inside famous landmarks (see examples of synthetic variants in Fig. 8 in Appendix). Ignoring a few obvious failure cases we discuss below, the resulting images are not completely unrealistic, *i.e.* time and weather changes are a specific case of scene illumination changes. We found that geometric consistency is often preserved after replacing an image with a synthetic variant (see Fig. 9 in Appendix).

**Failure cases for synthesis.** We observed a number of failures in the image generation process, the majority for indoor images, as expected (see some such cases in the middle and bottom examples of Fig. 8 in the Appendix). From Tab. 1, we see that keeping such images during training does not significantly hurt the model's performance. We believe this is because, even in such cases, part of the instance (*e.g.* the door in the middle example) is still visible, even in the most distorted variants.

**Limitations.** As discussed by Sarıyıldız et al. (2023), using generic generative models for specific vision tasks can be seen as performing targeted distillation given a set of textual prompts. This means

that success is generally bounded by the expressivity of the generative models and their ability to generate useful alterations for the provided textual prompts.

## 5 RELATED WORK

We discuss here the most related works. A broader discussion can be found in Appendix Sec. D.

**Synthetic data for improved visual understanding.** Most computer vision tasks are expected to be resilient to some image appearance variations, *e.g.* day to night, weather, and seasonal changes. This is typically handled with data augmentation (Hendrycks et al., 2020; Yun et al., 2019) and sometimes with more complex physical models. The latter synthetically modify the training images towards specific targeted weather or time of day conditions. They for instance add synthetic fog (Kahraman & de Charette, 2017; Sakaridis et al., 2018), rain (Halder et al., 2019; Tremblay et al., 2021; Hu et al., 2019) or darken images (Lengyel et al., 2021). Another strategy is to remove weather-related artifacts such as snow (Ren et al., 2017), raindrop (Qian et al., 2018; You et al., 2016), haze (Cai et al., 2017; Li et al., 2017; 2018a; Ren et al., 2018; Zhang & Patel, 2018), or darkness (Jenicek & Chum, 2019; Wu et al., 2021; Mohwald et al., 2023). Closer to our work, Fahes et al. (2023) uses a textual description of the target domain to perform domain adaptation for segmentation.

With the recent success of generative models, producing (Rombach et al., 2022; Zhang et al., 2023; Saharia et al., 2022) and modifying (Brooks et al., 2023; Nichol et al., 2022; Mokady et al., 2023; Parmar et al., 2023; Dunlap et al., 2023) images became within reach of even inexperienced users who can simply describe the desired content or changes in natural language. Such tools have been used to extend (Dunlap et al., 2023; Trabucco et al., 2023) or even replace (He et al., 2023; Azizi et al., 2023; Sarıyıldız et al., 2023) training image collections. We position our work on the continuity of this line of research, and extend our dataset using textual prompts relevant to localization.

**Visual localization under challenging conditions.** Image retrieval is often used as an initial step for structure-based visual localization (Humenberger et al., 2022) as it limits the search range in the 3D maps (Humenberger et al., 2020; Taira et al., 2021; Sarlin et al., 2019) and it even provides an approximate localization that is sufficient for place recognition (Zamir et al., 2016; Lowry et al., 2016; Berton et al., 2022). However, visual localization methods, including their retrieval component, need to be resilient to image appearance changes due to day to night changes as well as weather and seasonal variations. This problem is called *long-term visual localization* (Sattler et al., 2018; von Stumberg et al., 2020; Toft et al., 2022). On top of standard data augmentation or photometric normalization (Jenicek & Chum, 2019) applied to retrieval models (Revaud et al., 2019a), domain adaptation is sometimes used. It considers a single domain shift, *e.g.* day-*vs.*-night (Anoosheh et al., 2019; Zheng et al., 2020), or several ones (Hu et al., 2021). Tailored approaches have also been proposed. Porav et al. (2019) learn weather-specific adapters while Xin et al. (2019); Larsson et al. (2019) build from aligned image pairs composed of the exact same scenes observed under different conditions. Such requirement is rarely possible in practice. In contrast, we show that a tailored augmentation of the training set using both simple image processing (Hendrycks et al., 2020) as well as generative models with simple textual prompts can be automated and greatly improves performance on long-term visual localization benchmarks (Sattler et al., 2018). Another way to improve robustness to such variations is to leverage semantics during pose approximation, *e.g.*, define semantic-aware detectors which favor reliable regions (Xue et al., 2023). These methods are orthogonal to any improvement to the retrieval step, such as the ones we propose in this paper.

## 6 CONCLUSIONS

In this paper we propose to harness the power of recent text-to-image generative models and direct it towards specific domain shifts that affect visual localization. Generative models are extremely well suited for this task. First, unlike many other vision tasks, most of the challenges impacting the model can be expressed in natural language. Second, the fact that we are provided with matching pairs for training can be used for automatic quality control. This is all the more important as generative models are known to fail in unpredictable manner. To our knowledge, this is the first time such generative models are used for producing and validating targeted scene alterations. Finally, it is worth noting that our contributions are not handcrafted towards localization, and hence we can envision applications to other vision tasks.

**Acknowledgments.** The authors would like to sincerely thank Martin Humenberger, Jerome Revaud and Giorgos Tolias for many helpful discussions, as well as Yohann Cabon for providing us with Kapture evaluation scripts and datasets.

**Ethics Statement.** Our method automatically generates and uses altered versions of the training set images for training retrieval models. Those altered variants are automatically produced by generative models, using textual prompts. Our prompts were explicitly targeting seasonal, weather and time of day changes and our extended dataset was tested only on academic benchmarks for this study. Yet, those generative models are not fully understood and may generate biased or even problematic images. They have been released with little information about their training data and process, and we have used them as is. Consequently, a deeper study is required before using any retrieval models of such kind in production. Moreover, we can already foresee undesirable consequences in a number of other scenarios, *e.g.* when prompts of a sensitive nature are used. In general, when using the output of a model in any application involving humans, a rigorous validation needs to be performed.

**Reproducibility Statement.** We have built our approach as much as possible on top of public codebases, data and models. We have provided the training hyperparameters (Appendix A) and as much experimental details as possible. We believe that all the information available allows to fully reproduce our experimental validation. Moreover, we plan to publicly release models that will enable reproducing all results presented in the paper.

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

## CONTENTS

## A   DETAILS ON IMPLEMENTATION AND EXPERIMENTAL PROTOCOL

### A.1   DATASETS

In the main paper, we evaluate Ret4Loc-HOW models on the following datasets:

**Aachen Day-Night v1.1** (Sattler et al., 2012; Zhang et al., 2021) is a small dataset of high-resolution images. It contains 6,697 mapping images taken from handheld cameras under daytime in the historical city center of Aachen, Germany. Meanwhile, its 1,015 test queries were taken with different mobile phones at day and at night. This dataset encapsulates the scenario of city-wide outdoor localization of users thanks to easily accessible phone photos. Evaluation is presented separately to day and night images, as the second scenario is more challenging due to day-night domain shift.

**Robotcar Seasons v1** (Maddern et al., 2017) consists of an outdoor dataset extracted in Oxford, UK, thanks to three cameras (left, right and rear views) rigged on top of a vehicle roaming around the city. Its 20,864 mapping images were captured during day time, under overcast weather, while its 11,934 queries were captured during both day and night, and under a variety of weather and seasonal conditions. Following Sattler et al. (2018), we report results separately on day and night queries, and only on rear view queries. These splits represent the same day-night shift seen on Aachen Day-Night on top of the domain shift related to the seasonal domain shift. We also present results on the more recent split named **Robotcar Seasons v2** (Toft et al., 2022), that mixes day and night queries and has a larger (26,121) mapping set. Since kapture does not provide off-the-shelf 3D map for this split, we evaluate our models on **v2** only for EWB pose approximation. On **v1** we report results for both EWB and accurate camera localization using the provided Global-SfM map.

**Extended CMU Seasons** (Badino et al., 2011; Toft et al., 2022) is based on the CMU Visual Localization Dataset (Badino et al., 2011), and contains over 100,000 images collected over a period of 12 months in Pittsburgh, USA, with two cameras rigged on top of an SUV. The version of the dataset proposed by Toft et al. (2022) contains 24 separate sequences used to build submaps, as the full dataset is too large to build a single 3D model. These 24 sequences are split into a validation set, composed of 10 sequences for which the query poses are made available, and a test set, composed of 14 sequences for which no query poses are available and performance can only be tested on the *Long Term Visual Localization Benchmark* server.[7] We present results on both splits, referred to as ECMU-val and ECMU-test respectively. Additionally, we present results grouped according to the queried environment (urban, suburban or park) as well as weather and seasonal conditions (sunny, foliage, snow, etc.) in Table 4.

**Gangnam Station B2** (Lee et al., 2021) is an indoor dataset collected in a floor of the Gangnam metro station in Seoul, Korea. It is composed of 4,518 mapping and 1,786 test queries, collected with both industrial and phone cameras rigged on a dedicated mapping device. Images were collected at a crowed closed space, so scene occlusion is one of its main challenges. In addition, the dataset contains many digital signage and platform screen doors that change appearance over time. As this dataset spans a smaller physical space than other localization datasets, its localization accuracy thresholds are lower: $1m/5°$, $0.25m/2°$, $0.1m/1°$, for low, mid, and high-level accuracy, respectively. More details about the localization metrics are presented in Appendix A.3.

**Baidu Mall dataset** (Sun et al., 2017) was collected in a shopping mall of over 5,000 square meters in China. The dataset is composed of 689 mapping images captured with high-resolution cameras on an empty mall and 2,292 phone camera images taken when the mall was open. This indoor dataset contains repetitive structures, reflective and transparent surfaces, and scene occlusion, which represent a source of challenges complementary to what we observe on outdoor localization.

**Tokyo-24/7** (Torii et al., 2015a) is a place recognition dataset that contains 6,332 geo-tagged street-view panoramas from Tokyo, Japan, which are then split into the 75,984 views we use as retrieval gallery. As queries, we use 315 images collected with phone cameras at 125 different locations and times.

**Pittsburgh-30k** (Arandjelović et al., 2016) is a place recognition dataset, subset of the larger Pittsburgh-250k dataset (Torii et al., 2013). Its test set contains 10,000 geo-tagged images downloaded from Google Street View, and 6,816 query images from the Google Pittsburgh Research Dataset. These queries are extracted from the same locations as the gallery images, but at different times and across several months.

## A.2 TRAINING DETAILS

We build on top of the HOW codebase[8] and mostly follow the protocol for training models from HOW, including hyper-parameters. More precisely, we finetune a ResNet50 model (He et al., 2016) pretrained on ImageNet-1K (Russakovsky et al., 2015). We remove the global average pooling and the last fully connected layers from the ResNet50 model, and attach $\ell_2$-norm based spatial pooling and dimensionality reduction (from 2048 dimensions to 128, obtained by PCA) layers to obtain the final representation of an image. We use "randomly resized crops" of size $768 \times 768$ with the default parameters from torchvision (Paszke et al., 2019) during training as additional data-augmentation, as well as to provide the network with fixed sized inputs. This improves the computational aspect of our approach especially when we use multiple synthetic variants of an image in the same batch during training.

On top of that, we apply AugMix (Hendrycks et al., 2020) data augmentations to random crops with its default hyper-parameters suggested by the authors:[9] severity of augmentation operators: 3, width of augmentation chain: 3, depth of augmentation chain: -1, probability coefficient for Beta and Dirichlet distributions: 1. The exception is the list of augmentations, which is appended by color, contrast, brightness and sharpness transformations. We also removed translation and shear, which do not preserve the geometry of an image's local features.

---

[7]https://www.visuallocalization.net/
[8]https://github.com/gtolias/how
[9]https://github.com/google-research/augmix

We perform "episodic" training on SfM-120K in the sense that in each episode, we sample 2000 random query-positive pairs and a pool of 20000 images to be used for hard negative mining. We train each model for 50 such episodes. Given that the size of the training set is close to 100K, our models barely see each image once during training. Thanks to using fixed-sized images as network input, we can adjust batch size to be proportional to the number of tuples during training. To normalize the impact of the number of tuples into gradients, we divide Eq. (2) by the number of tuples used in the batch. Within the limits of our compute infrastructure, using 5 tuples, where each tuple consists of 1 query, 1 positive and 5 negative images, gives the best performance. We used $M = 5$ negatives in the tuple as this is the default setting used by HOW. We set the learning rate and weight decay arguments for the AdamW optimizer for each model configuration separately (*e.g.* for Ret4Loc-HOW, Ret4Loc-HOW-**Synth**, *etc.*), however, learning rate 1e-5 and weight decay 3e-2 work the best in many cases.

**Issues with variance.** We observe large variance in performance when training with the exact same setup but different seeds. This is also observed up to some degree when training the baseline HOW. We therefore choose to report results for each setup after averaging the weights (Wortsman et al., 2022) of 3 models, *i.e.* obtained with the exact same setup starting from 3 different seeds.

**Retrieval at test-time.** We use the ASMK implementation[10] that further uses the FAISS library (Johnson et al., 2019)[11] for sub-linear indexing. We use 3000 local features for matching and learn the ASMK codebook on SfM-120k. We follow the protocol of HOW/FIRe and perform multi-scale testing over 7 scales.

## A.3 METRICS AND EVALUATION PROTOCOLS

In this section, we further describe the evaluation protocol we briefly mentioned in Section 3 of the main paper. As mentioned, our retrieval models output, for each query image, a list of the $k$ nearest neighbors among the mapping images, for which we know the camera pose. For EWB, the approximated pose is simply obtained by an arithmetic mean of the poses of these mapping images. This protocol is very light-weight, and ideal for real-time localization scenarios. Its drawback is the lack of any geometric verification for the proposed pose. The quality of the approximated pose also depends if images taken from similar poses exist amongst the mapping images. Indeed, EWB for $k = 1$ is equivalent of approximating the query pose by the pose of its most similar mapping image. As we increase the value of $k$, performance tends to fall as we account for more distant similar images or with different viewpoints, yielding in general a less accurate barycenter.

For Global-SfM, we extract local features for all retrieved $k$ mapping images and feed together with the query image to the kapture-localization pipeline. As these mapping images are from the ones used to compute the SfM model of the scene (in our case with COLMAP (Schönberger & Frahm, 2016)), the SfM model contains 3D correspondences to 2D local features for these mapping images. Hence, once we establish 2D-2D matching of local features between the query image and the top ranked mapping images, we obtain a set of 2D-3D matches between the 3D scene representation and the query image which are used for the query pose estimation via PNP and RANSAC.

In our experiments, both the local features and the COLMAP are obtained off-the-shelf from kapture[12].They both rely on the 'featheR2D2_32_20k' local features, — a variant of R2D2 (Revaud et al., 2019b) — which consists of 20K 32 dimensional features extracted per image. In contrast to EWB, the Global-SfM protocol obtains better accuracy as we increase the value of k, since more relative poses can be taken into account and used for triangulation to estimate the query pose. On the other hand, larger values of $k$ also increase the computational cost of the protocol, as local feature matching is often the main computational bottleneck. Therefore, for practical applications, the value of $k$ should be selected such that it provides good pose estimation within a time budget.

For all protocols, we measure localization performance by the percentage of correctly estimated camera poses withing a given error threshold. Indeed, given the groundtruth and the estimated positions, we calculate the translation and rotation differences between the two positions (Humenberger et al., 2022), and we consider an image as successfully localized if the translation error is below X

---

[10]https://github.com/jenicek/asmk
[11]https://github.com/facebookresearch/faiss
[12]https://github.com/naver/kapture

meters and the rotation error is below Y°. We report the percentage of localized queries at three levels of accuracy: low-level (X=5m, Y=10°), mid-level (X=0.5m, Y=5°) and high-level (X=0.25m, Y=2°) accuracy[13]. We prioritize showing different levels of accuracy for different protocols: for EWB, as a first-level approximation, we show low-level accuracy, (in many cases, depending on the mapping images, the high accuracy level is not achievable even with an oracle), whereas for Global-SfM, the target is precise pose estimation, therefore we prioritize mid and high-level accuracy.

For place recognition, we follow Torii et al. (2015a) and consider that two images represent the same place if their camera positions are at most 25 meters apart. We compare performance by Top-k recall (R@k), which measures the percentage of queries for which at least one of the k nearest neighbors represents the same place as the query.

## A.4 BASELINE RETRIEVAL MODELS EVALUATED ON LOCALIZATION

For fairness, we evaluated ourselves some of the top existing retrieval models used for visual localization under the exact same setup we use for the proposed Ret4Loc models. We evaluated the following models under the Kapture framework (Humenberger et al., 2020):

- **NetVLAD**[14] (Arandjelović et al., 2016) is a model based on a generalized VLAD layer, that aggregates local convolutional features extracted with a VGG-16 (Simonyan & Zisserman, 2015) backbone, and learned end-to-end with a triplet loss. The model is trained on geo-tagged data collected at different camera angles, lighting conditions and seasons. NetVLAD is a milestone work in visual place recognition and it is used as baseline for different methods in the task (Hausler et al., 2021; Leyva-Vallina et al., 2023), and it has been used in state-of-the-art visual localization pipelines (Germain et al., 2019; Sarlin et al., 2019). We report results with a model trained on Pittsburgh-250k (Torii et al., 2013), which outputs a global representation of 4096 dimensions, from the PyTorch re-implementation of NetVLAD from Hausler et al. (2021).

- **AP-GeM**[15] (Revaud et al., 2019a) is a model trained for image retrieval with a differentiable approximation of the average precision (AP) metric, which is the main evaluation metric used in landmark retrieval, which learns a global image representation obtained from the generalized-mean pooling (GeM pooling) (Radenović et al., 2019) of local convolutional features of a ResNet (He et al., 2016). We follow Humenberger et al. (2020), who shows that AP-GeM is a strong retrieval model for visual localization, and use the model trained on Google Landmark dataset v1 (GLD) (Noh et al., 2017).

- **HOW**[16] (Tolias et al., 2020) is the retrieval model we use as bases for our Ret4Loc models. HOW is trained on global image representations, obtained from aggregating intermediary convolutional features with SPoC pooling (Babenko & Lempitsky, 2015), with a contrastive margin loss over matching image pairs. At inference time, HOW local features are matched with ASMK (Tolias et al., 2013) for improved performance. We use the model released by the authors as baseline.

- **FIRe**[17] (Weinzaepfel et al., 2022) is a retrieval model that builds upon HOW's formula of training with a global representation while performing inference with matching lower-level representations. However, instead of aggregating local convolutional features, it uses an attention module to propose mid-level *super-features* for matching. Weinzaepfel et al. (2022) also first benchmarked HOW and FIRe for visual localization on Aachen, hinting to their potential as state-of-the-art methods for the task. For both HOW/FIRe, we use the ResNet50-based model released by the authors, trained on SfM-120k (Radenović et al., 2019), and follow the same test-time protocol as for our models, detailed in Appendix A.2.

---

[13]As mentioned above, for Gangnam Station the thresholds are 1m/5°, 0.25m/2°, 0.1m/1°, for low, mid, and high-level accuracy

[14]We use the re-implementation from https://github.com/QVPR/Patch-NetVLAD

[15]https://github.com/naver/deep-image-retrieval

[16]https://github.com/gtolias/how

[17]https://github.com/naver/fire

A.5    RE-RANKING EXPERIMENTS

In Tab. 2 and Tab. 5 we also report results after geometric re-ranking. For each query, we first obtain ranked lists using our Ret4Loc-HOW-**Synth++** model, and match the query image to each of the top 100 closest images using LightGlue (Lindenberger et al., 2023). We then re-rank the top images according to the number of inliers.

We use the public LightGlue implementation[18] with SuperPoint (DeTone et al., 2018) features. We extract at most 3072 keypoint features per image, with all images of a dataset resized to a fixed resolution for consistency (1024 pixels as the largest dimension for Tokyo 24/7, RobotCar, and Extended CMU, and 750 pixels for Pittsburgh30k). In order to maximize accuracy, we disable early stopping and iterative point pruning.

B    FURTHER EXPERIMENTAL ANALYSES AND RESULTS

B.1    STATISTICS FOR THE FILTERING THRESHOLD $\tau$

Here, we study the effect of the selection threshold $\tau$.

In Fig. 4 we plot on the left the percentage of synthetic pairs that would be dropped for different thresholds $\tau$ on the geometric consistency score $s$. On the right we report the percentage of synthetic pairs that are dropped for $\tau = 0.5$ for each textual prompt. We see that 'at sunset' has a much higher probability of being dropped; the strong colors of sunset images dominate after alteration in many cases, making geometric consistency fail.

In Fig. 5 we study the effect of the selection threshold and report gains on RobotCar-v2 for different values of $\tau$ with and without geometric-aware sampling. We see that performance is generally consistent for $\tau = [0.2, 0.3]$ and slightly dropping for $\tau = 0.5$. We select the best value of $\tau$ among $\{0.2, 0.3\}$ via a validation set. For the two named Ret4Loc models, Ret4Loc-HOW-**Synth+** uses $\tau = 0.3$ and Ret4Loc-HOW-**Synth++** uses $\tau = 0.2$.

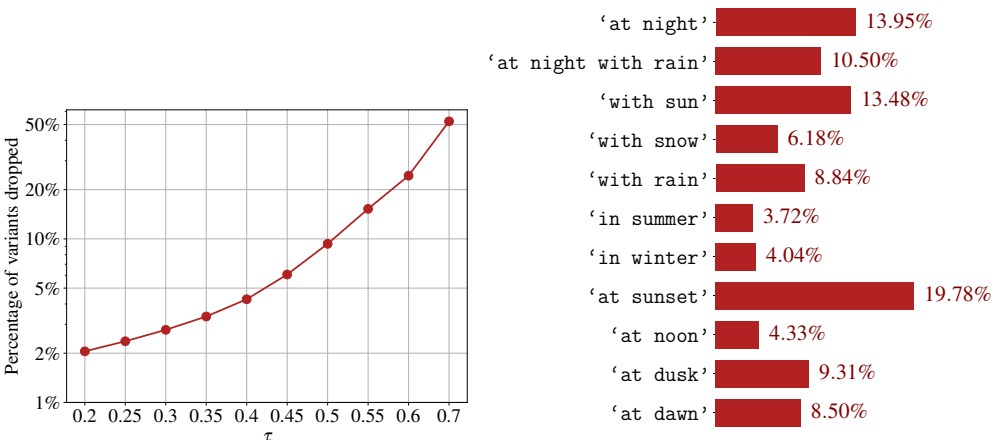

(a) Synthetic pairs dropped for $0.2 \geq \tau \geq 0.7$.    (b) Synthetic pairs dropped per prompt for $\tau = 0.5$.

Figure 4: (Left) Percentage of synthetic pairs dropped for different thresholds $\tau$ on the geometric consistency score $s$. (Right) Percentage of synthetic pairs dropped per textual prompt for $\tau = 0.5$.

B.2    COMPLETE SETS OF RESULTS FOR FIVE COMMON DATASETS

In Figs. 10 to 12 we show complete results for 5 popular long-term visual localization datasets available on the Visual Localization Benchmark[19] and available in the kapture platform (Humenberger

---

[18]https://github.com/cvg/LightGlue
[19]https://www.visuallocalization.net/

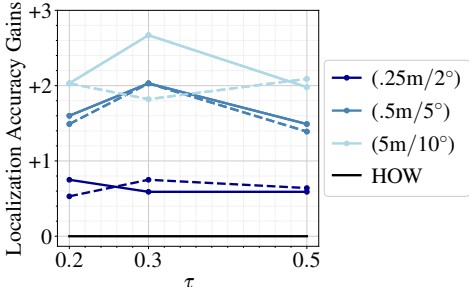

Figure 5: **Relative gains on Robotcar-v2** for different $\tau$ values for Ret4Loc-HOW-**Synth+** (solid lines) and Ret4Loc-HOW-**Synth++** (dashed lines) for localization across three error thresholds.

et al., 2020), *i.e.* Aachen Day-Night, Robotcar Seasons, Extended CMU Seasons, Baidu Mall and Gangnam Station B2. We report results for the highest error threshold for EWB and the two lowest for SfM (we explain this choice in Appendix A).

### B.3    RESULTS PER QUERY CONDITION FOR ECMU-TEST

We present in Tab. 4 the in-depth results for our proposed models and some baseline on the Extended CMU Seasons test split. The results for this split are only available via submission to the Long-Term Visual Localization benchmark[20], and it provides performance broken down according to query environment (urban, suburban and park) as well as seasonal or weather conditions. Firstly, we notice that our Ret4Loc-HOW models outperform HOW on all conditions, with improvements superior to 3% on several query types where HOW struggles (*i.e.* accuracy below 90%), such as park, sunny, foliage, and mixed foliage. Our base Ret4Loc-HOW present the best results on overcast, sunny, and foliage, which validates the importance of the changes our method brought to HOW, namely the optimization and the necessity of data augmentations. Secondly, Ret4Loc-HOW+synthetic performance on rows 3 and 4 shows straight improvements to some types of queries with respect to our base Ret4Loc-HOW, in particular to suburban, no foliage, and snow queries. Such conditions seem to be better addressed with synthetic images instead of pixel-level data augmentations. However, such improvements are not generalized, as overcast, sunny, foliage and cloudy see their results drop with respect to Ret4Loc-HOW. These set-backs are finally addressed with the use of geometry-aware sampling, as we notice in rows 5 and 6, showing the importance of the geometric consistency based sampling and filtering. Ret4Loc-HOW+synthetic models in these lines obtain the best results for all but two conditions, and obtain the best overall results on the Extended CMU Seasons test.

| | | | | | | | ECMU-Test, EWB (top-1), 5m/10° | | | | | | | | |
|---|---|---|---|---|---|---|---|---|---|---|---|---|---|---|---|
| Model | VMix | GaS | Filt | Urban | Sub-urban | Park | Overcast | Sunny | Foliage | Mixed Foliage | No Foliage | Low Sun | Cloudy | Snow | All |
| 1. HOW | – | – | – | 94.4 | 88.8 | 79.0 | 87.2 | 83.1 | 83.9 | 88.7 | 95.1 | 90.0 | 90.9 | 91.7 | 87.9 |
| 2. Ret4Loc-HOW | – | – | – | 95.4 | 89.5 | 86.0 | **90.0** | **86.7** | **87.3** | 91.4 | 95.6 | 92.0 | 93.0 | 92.9 | 90.5 |
| 3. | – | – | – | 95.3 | 89.9 | 81.8 | 88.9 | 84.2 | 85.2 | 90.4 | 96.6 | 91.9 | 92.3 | 94.6 | 89.4 |
| 4. | ✔ | – | – | 95.2 | 90.4 | 86.1 | 89.7 | 86.3 | 86.9 | 92.2 | 96.7 | 93.2 | 93.0 | 94.9 | 90.8 |
| Ret4Loc-HOW + synthetic | | | *Variations using geometric consistency* | | | | | | | | | | | | |
| 5. | ✔ | ✔ | – | 95.4 | **90.7** | 86.1 | **90.0** | 86.3 | 86.9 | 92.5 | **97.2** | 93.7 | 93.0 | **95.7** | 91.0 |
| 6. | ✔ | ✔ | ✔ | **95.5** | **90.7** | **86.7** | 89.9 | 86.5 | 87.1 | **92.8** | 97.0 | **93.9** | 93.4 | 95.4 | **91.2** |

Table 4: **Extended CMU Seasons test results broken down by query conditions. Bold** number represent the best results per column. *VMix* refers to the use of variant mixing with Eq. (2) instead of simple loss averaging, *Filt.* to synthetic tuple filtering, *GaS* to geometry-aware sampling.

---

[20] https://www.visuallocalization.net/

| | Model | Tokyo 24/7 | | | Pitts30k | | |
|---|---|---|---|---|---|---|---|
| | | Retrieval (top-$k$ recall) | | | Retrieval (top-$k$ recall) | | |
| | | R@1 | R@5 | R@10 | R@1 | R@5 | R@10 |
| | *Single stage retrieval methods* | | | | | | |
| 1. | NetVLAD* | 37.8 | 53.3 | 61.0 | 70.3 | 84.1 | 89.1 |
| 2. | TransVPR* | - | - | - | 73.8 | 88.1 | 91.9 |
| 3. | GCL* | 69.5 | 81.0 | 85.1 | 80.7 | 91.5 | 93.9 |
| 4. | FIRe | 84.8 | 89.8 | 92.7 | 84.5 | 92.2 | 94.8 |
| 5. | HOW | 89.2 | 94.6 | 96.5 | 84.4 | 92.0 | 94.6 |
| 6. | Ret4Loc-HOW | 89.2 | **95.6** | **97.1** | 85.0 | 92.6 | **94.9** |
| 7. | Ret4Loc-HOW-**Synth++** | **91.1** | 94.0 | **97.1** | **85.9** | **92.9** | **94.9** |
| | *Retrieval methods **with re-ranking*** | | | | | | |
| 8. | SP-SuperGlue* | 88.2 | 90.2 | 90.2 | 87.2 | 94.8 | 96.4 |
| 9. | DELG* | 95.9 | 96.8 | 97.1 | **89.9** | **95.4** | **96.7** |
| 10. | Patch NetVLAD* | 86.0 | 88.6 | 90.5 | 88.7 | 94.5 | 95.9 |
| 11. | TransVPR* | - | - | - | 89.0 | 94.9 | 96.2 |
| 12. | Ret4Loc-HOW-**Synth++** | **97.5** | **98.1** | **98.4** | 88.5 | 93.8 | 96.2 |

Table 5: **Visual place recognition results on place recognition for Tokyo 24/7 and Pittsburgh30k.** We report the usual metrics (top-$k$ recall), *i.e.* if one correct image is retrieved in the top $k$. * denotes results from GCL (Leyva-Vallina et al., 2023);

### B.4 RETRIEVAL PERFORMANCE ON VISUAL PLACE RECOGNITION DATASETS

In Tab. 5 we present results on the Tokyo 24/7 and Pittsburgh30k place recognition datasets. We see that our models consistently outperform compared methods for the case of single-stage retrieval, while, when paired with SuperPoint+LightGlue re-ranking (see Appendix A.5 for details), Ret4Loc models achieve a new state-of-the-art for Tokyo 24/7 dataset, a commonly used place recognition dataset that focuses on weather and seasonal changes.

### B.5 FEATURE SPACE ANALYSIS

In order to better understand how individual model components impact representations, we analyze features learned by the baseline HOW model and 4 of our models: Ret4Loc-HOW, Ret4Loc-HOW-**Synth**, Ret4Loc-HOW-**Synth+** and Ret4Loc-HOW-**Synth++**.

To this end, we sample 1000 real images from the validation set of SfM-120K and extract features for each of those images along with their 11 synthetic variants (which we generate as described in Sec. 2.2). While extracting features, we resize images such that their longest side is 1024 pixels, and we do not apply any cropping to images. In total, we obtain 12000 128-dimensional image features for each model. Then, we create 1000 "classes" by grouping each real image with its synthetic variants and assign a pseudo-label to each such pseudo-class. These pseudo-labels are used for ground-truth in the following analyses we perform.

**Alignment *vs*. uniformity analysis.** Following Wang & Isola (2020),[21] we measure the alignment loss between the feature of a real image and each of its 11 synthetic variants, and the uniformity loss across all features. The results are shown in Fig. 6. We see that HOW and Ret4Loc-HOW are located on the upper left and lower right corners, minimizing either the uniformity or alignment losses, respectively. Whereas, our 3 models trained with synthetic data (Ret4Loc-HOW-**Synth**, Ret4Loc-HOW-**Synth+** and Ret4Loc-HOW-**Synth++**) exhibit a better balance in the trade-off between the two losses, positioning closer to the lower left corner. We note that this analysis was originally performed in the context of visual representation learning via self-supervised models, and Wang & Isola (2020) observed that best performing models tend to position towards the bottom left part of such 2D plots, as indicated by darker blue in Fig. 6.

**Clustering analysis.** We cluster all 12000 features for each model into 1000 clusters via $k$-means. Then we measure the purity, adjusted random index (ARI) and normalized mutual information

---
[21]https://github.com/SsnL/align_uniform

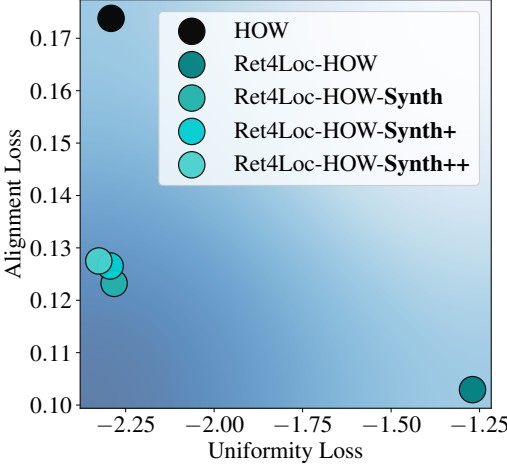

Figure 6: **Alignment *vs*. uniformity analysis** (Wang & Isola, 2020) using pre-extracted features from the validation set of SfM-120K. We sample 1000 real images from the validation set and extract features for each of those images along with their 11 synthetic variants (which we generate as described in Sec. 2.2). Each real and its 11 synthetic variants form a "class", which we use to measure the alignment loss. More concretely, we measure the alignment loss between a real image and each of its 11 synthetic variants. The uniformity loss is measured across all features. This analysis was originally performed in the context of visual representation learning via self-supervised models, and Wang & Isola (2020) observed that best performing models tend to position towards the bottom left part of such 2D plots, as indicated by darker blue in the figure.

| Method | $K$ | training images (total) | training time | max. GPU memory |
|---|---|---|---|---|
| Ret4Loc-HOW | – | 91K (real) | ∼4 hours | 21.8 GB |
| Ret4Loc-HOW-**Synth++** | 3 | 91K (real) + 1M (synthetic) | ∼8 hours | 65.4 GB |

Table 6: **Training time and memory usage for Ret4Loc models.** Note that these timings *do not include the time needed for synthetic data generation*, since we ran generation as a pre-processing step, only once. Timings for image generation are presented in Tab. 7.

(NMI) scores between hard cluster assignments and pseudo-labels. The results are reported in Tab. 3. We observe that our models trained with synthetic data consistently improve the clustering metrics over HOW and Ret4Loc-HOW.

### B.6 GENERATION AND TRAINING TIMES

In this section we separately report and discuss i) the model training time and memory usage, and ii) the synthetic image generation time, as, in practice, we ran synthetic image generation *as an offline process* once and saved all 11 variants for each training image.

Tab. 6 reports training time and memory usage for Ret4Loc model obtained using a single A100 GPU. Again, training times in this table do not include time corresponding to the image generation process.

We present synthetic image generation time in Tab. 7. The latter is affected by two parameters: the generated image resolution (Resolution) and the number of diffusion steps (Steps). The table shows the generation time per image in seconds (Time), for different resolutions and number of steps. We chose to use a resolution of 768 pixels and 20 steps for all results in the paper. For these parameters, the time needed for generating all synthetic images for a single prompt was approximately 25 hours for the complete SfM-120k dataset. Given that the process is independent per prompt, we ran synthetic data extraction in parallel for all prompts and generated images for all variants *in about a day*. With model training taking approximately 8 hours, we see that data generation time is at least 3

| Resolution | Steps | Time (sec) |
|---|---|---|
| 512 | 10 | 0.30 |
| 512 | 20 | 0.44 |
| 512 | 30 | 0.64 |
| 768 | 10 | 0.62 |
| 768 | 20 | 0.98 |
| 768 | 30 | 1.39 |
| 1024 | 10 | 1.16 |
| 1024 | 20 | 1.99 |
| 1024 | 30 | 2.78 |

Table 7: **Generation time for InstructPix2Pix.** Time required to generating a single variant for an image, hence approximately 25 hours for processing the full SfM-120k dataset (91K images) for resolution 768 and 20 steps. Different variants can be processed in parallel.

times higher than the actual model training. However, we do not believe this is an issue in practical applications for mainly three reasons:

1. As a pre-processing step, data generation needs to run only once. This is unlike model training, a process that one usually needs to run multiple times, *e.g.* to perform hyperparameter validation.

2. The overall training time for our model (data generation + model training) is approximately 32 hours. This means that learning state-of-the-art retrieval models for visual localization using our method is manageable in reasonable time, even with modest resources.

3. We believe that over time the data synthesis overhead will eventually be even less of an issue: generative AI models will inevitably become more efficient and we envision in the near future such generation to happen on-the-fly, during batch construction.

## C QUALITATIVE EVALUATION OF SYNTHETIC VARIANTS

In Fig. 7, we show the complete set of 11 variants for the three images depicted in Fig. 2a.

### C.1 RESULTS WITH CONTROLNET

When exploring which generative AI model to use, besides InstructPix2Pix, we also tested ControlNet (Zhang et al., 2023). We found the latter to produce images far more stylized and we deemed them unfit for our purpose. We compare InstructPix2Pix and ControlNet generations for the images depicted in Fig. 2a of the main paper in Fig. 14 and Fig. 15.

### C.2 GEOMETRIC CONSISTENCY FOR DIFFERENT STEPS OF THE GENERATOR

In Fig. 13 we show geometric consistency results when varying the number of diffusion steps, *i.e.* using 2, 10, 20 or 30. We see that 10 steps are enough for getting a large number of geometric matches. After preliminary explorations, we set the number of steps for generation to 20 for all results presented in the paper.

### C.3 SYNTHESIS FAILURE CASES

In Fig. 8, we show what synthetic variants look like for cases where most prompts don't really make sense, *i.e.* for indoor images. We see that in many cases generations are still plausible and can be interpreted as extreme illumination changes. We can also see a few failure cases in the middle ("at dawn/dusk/sunset") and bottom ("with sun") examples in Fig. 8. As we show in Tab. 1, keeping such images during training does not really decrease the model's performance significantly. We believe this to be because, even in such cases, a part of the depicted instance (*e.g.* the door in the middle example or parts of the walls in the bottom one) is still visible even in the most distorted variants.

### C.4 GEOMETRIC CONSISTENCY FOR INDOOR AND FAILURE CASES

In Fig. 9, we show some interesting cases of geometric matching for a failure case (left) and for an indoor image. We see that even some of the extremely distorted images (left side, third row) or images with unrealistic alterations, *i.e.* the indoor image on the right altered with an "at sunset" prompt, can still be successfully matched with the LightGlue (Lindenberger et al., 2023) method.

## D EXTENSION OF THE RELATED WORK

In this section, we mention additional prior work that we consider related to our submission and that could provide a broader perspective. It complements the discussion of the most related works available in Sec 5.

**Retrieval for visual localization and place recognition.** Visual localization consists in estimating the 6-DoF camera pose from a single RGB image within a given area. The reference area can be represented as a 3D point cloud, also referred to as an SfM map (Se et al., 2002; Irschara et al., 2009; Li et al., 2010; Sattler et al., 2017; Sarlin et al., 2019; Taira et al., 2021; Germain et al., 2019; Humenberger et al., 2020) and camera localization relies on using 2D-3D matches between a query image and the 3D representation. Alternatively, learning-based localization methods are trained to regress 2D-3D matches (Shotton et al., 2013; Brachmann & Rother, 2019; Cavallari et al., 2017; Brachmann & Rother, 2022) or to directly predict the camera pose (Kendall et al., 2015; Sattler et al., 2019). State-of-the-art visual localization approaches often rely on image retrieval techniques to either provide an approximate pose estimate that is particularly relevant for place recognition known also as geo-localization (Zamir et al., 2016; Kim et al., 2017; Lowry et al., 2016; Leyva-Vallina et al., 2023; Torii et al., 2015b; 2018; Arandjelović et al., 2016), or as an initial step for SfM based methods as it allows large scale deployment. Indeed, it was shown that a good retrieval model not only reduces the localization computational cost by limiting the search to the scene parts potentially visible in a given query image, but can also yield to improved localization accuracy (Humenberger et al., 2020; Taira et al., 2021; Sarlin et al., 2019).

Typically, visual localization methods that rely on compact image-level descriptors in their first retrieval step use representations trained for landmark retrieval or place recognition (Torii et al., 2018; Arandjelović et al., 2016; Liu et al., 2019; Radenović et al., 2019; Revaud et al., 2019a; Kim et al., 2017; Leyva-Vallina et al., 2023). In order to further refine the retrieval step used in localization, several works propose to apply local feature based re-ranking of the top retrieved images (Sarlin et al., 2019; Wang et al., 2022; Sarlin et al., 2020; Cao et al., 2020; Hausler et al., 2021). Moreover, local features have been employed without aggregation for retrieval for a long time. Among them, the most recent ones rely on Aggregated Selective Match Kernels (ASMK) (Tolias et al., 2013). This is the case of HOW (Tolias et al., 2020) and FIRe (Weinzaepfel et al., 2022). In a concurrent work, Aiger et al. (2023) replace the ASMK matching step with Constrained Approximate Nearest Neighbors (CANN) and shows large gains for HOW and FIRe. In our approach, which also relies on HOW, we could also replace ASMK with CANN to further improve the results, but we consider this to be out of the scope of this paper.

**Building datasets for training models robust to adversarial weather, seasonal and lighting conditions.** The recent progress on computer graphics platforms enables the generation of photo-realistic virtual worlds with diverse, realistic, and physically plausible conditions including adversarial weather, seasonal and lighting conditions (Ros et al., 2016; Richter et al., 2016; Gaidon et al., 2016; Cabon et al., 2020). However there are two issues with such platforms. First, the generation of such datasets still requires a large amount of expert knowledge and important manual effort. Second, while they might help reducing the domain gap between these conditions, they might introduce an extra domain shift, known as the sim-to-real gap. Therefore, in parallel, researchers also collected real images sometimes recorded from the same streets at different times and under different conditions (Maddern et al., 2017; Toft et al., 2022; Sakaridis et al., 2019; 2018), to be used either to properly assess the impact of adverse weather/daylight on the models or to train models robust to such conditions. A third option to generate such datasets is to add weather-related artifacts on top of existing images (Ren et al., 2017; Qian et al., 2018; You et al., 2016; Cai et al., 2017; Li et al., 2017; 2018a; Ren et al., 2018; Zhang & Patel, 2018; Jenicek & Chum, 2019; Wu et al., 2021).

With the recent success of image generative models (Rombach et al., 2022; Zhang et al., 2023; Qin et al., 2023; Brooks et al., 2023; Nichol et al., 2022), generating photo-realistic images became much simpler. User without any expertise can easily augment existing datasets or even generate completely synthetic image sets (Dunlap et al., 2023; Trabucco et al., 2023; He et al., 2023; Azizi et al., 2023; Sarıyıldız et al., 2023). However, our augmented SfM-120K dataset (Radenović et al., 2019), is the first one that focuses on adversarial weather, seasonal and lighting conditions with the aim of improving long-term visual localization.

**Leveraging semantics to learn features robust to weather and seasonal conditions.** To address robustness to weather or seasonal variations, one could alternatively leverage semantics as an auxiliary source of information and learn robust representations that encapsulate some high-level semantic information, so they are by design more robust to appearance changes (Kobyshev et al., 2014; Toft et al., 2017; Schönberger et al., 2018; Toft et al., 2018; Garg et al., 2018; Yu et al., 2018; Shi et al., 2019; Benbihi et al., 2020; Hu et al., 2021; Paolicelli et al., 2022; Xue et al., 2023). All these works nevertheless require to train an additional full network, or at least an additional segmentation head, to be able predict the semantic information. On top of the extra training cost, these additional components also need to be trained in a way that is robust to domain shifts and dataset variations. This is an non trivial goal which constitutes an full research field in itself (Csurka et al., 2022).

**Visual style transfer.** Visual style transfer allows to combine the content of one image with the style of another one to create a new image. Such techniques have been used to mitigate the domain shift that can exist between training and test images, in particular could also handle visual appearance variations, due to seasonal, weather, or daylight changes. By exploiting the progress in image-to-image translation and style transfer (Gatys et al., 2015; Huang & Belongie, 2017; Li et al., 2018b), several methods have used them either as a pre-processing step or to align images (Csurka et al., 2017; Thomas & Kovashka, 2019; Melekhov et al., 2021), to learn to synthesize target-like images (Liu et al., 2017; Romera et al., 2019; Xu et al., 2021), independently or integrated within domain adaptation techniques (Hoffman et al., 2018; Wu et al., 2018; Li et al., 2019; Choi et al., 2019; Toldo et al., 2020; Cheng et al., 2021; Wang et al., 2021; Yang et al., 2021). In general, these models learn to cope with new appearances/styles that are learned from a set of images representing the targeted style or appearance. Furthermore, while in theory applicable to our problem, few of them addressed variations due to seasonal, weather, or daylight changes. In contrast, our model changes the appearance of the training images where the new appearance, in particular adversarial weather and daylight variation is simply described with natural language.

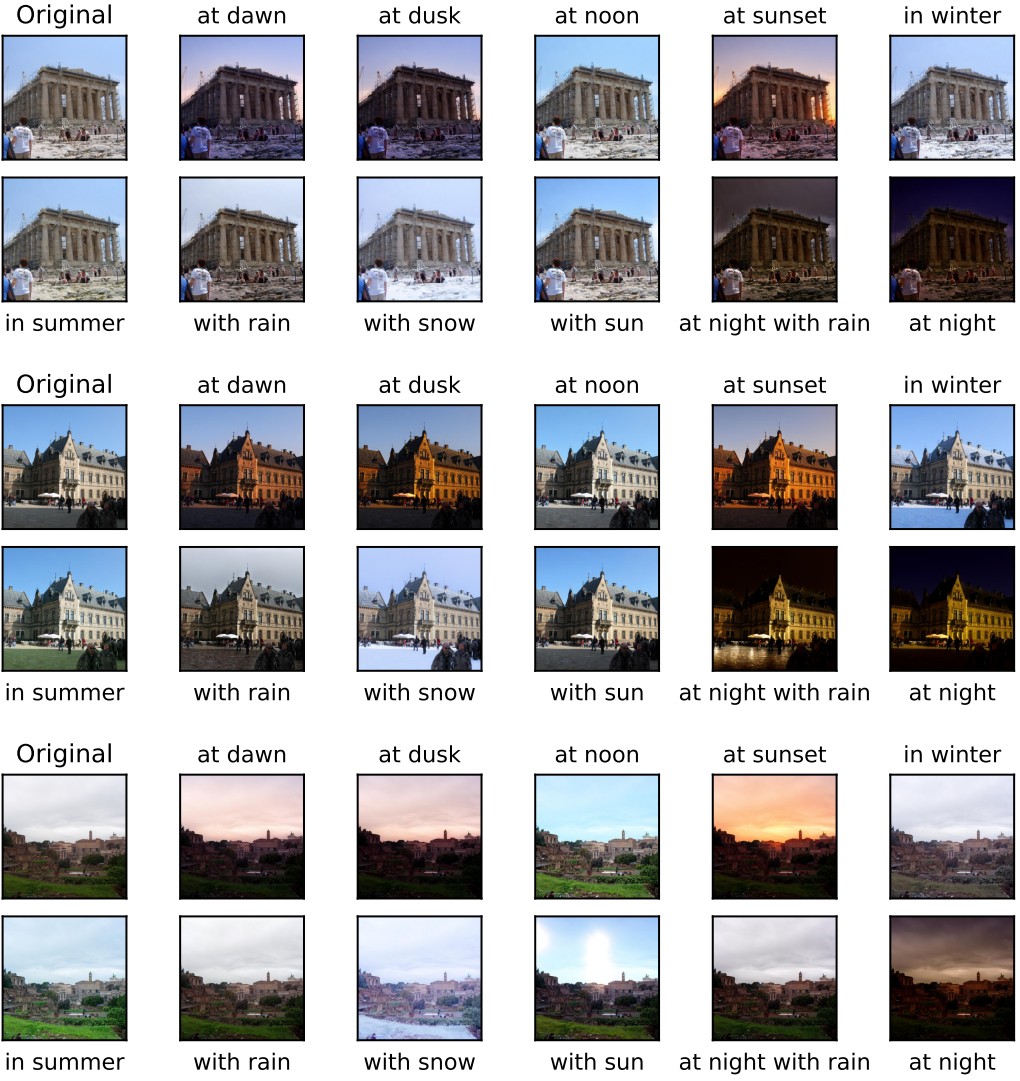

Figure 7: **All synthetic variants** for training images shown in Fig. 2a.

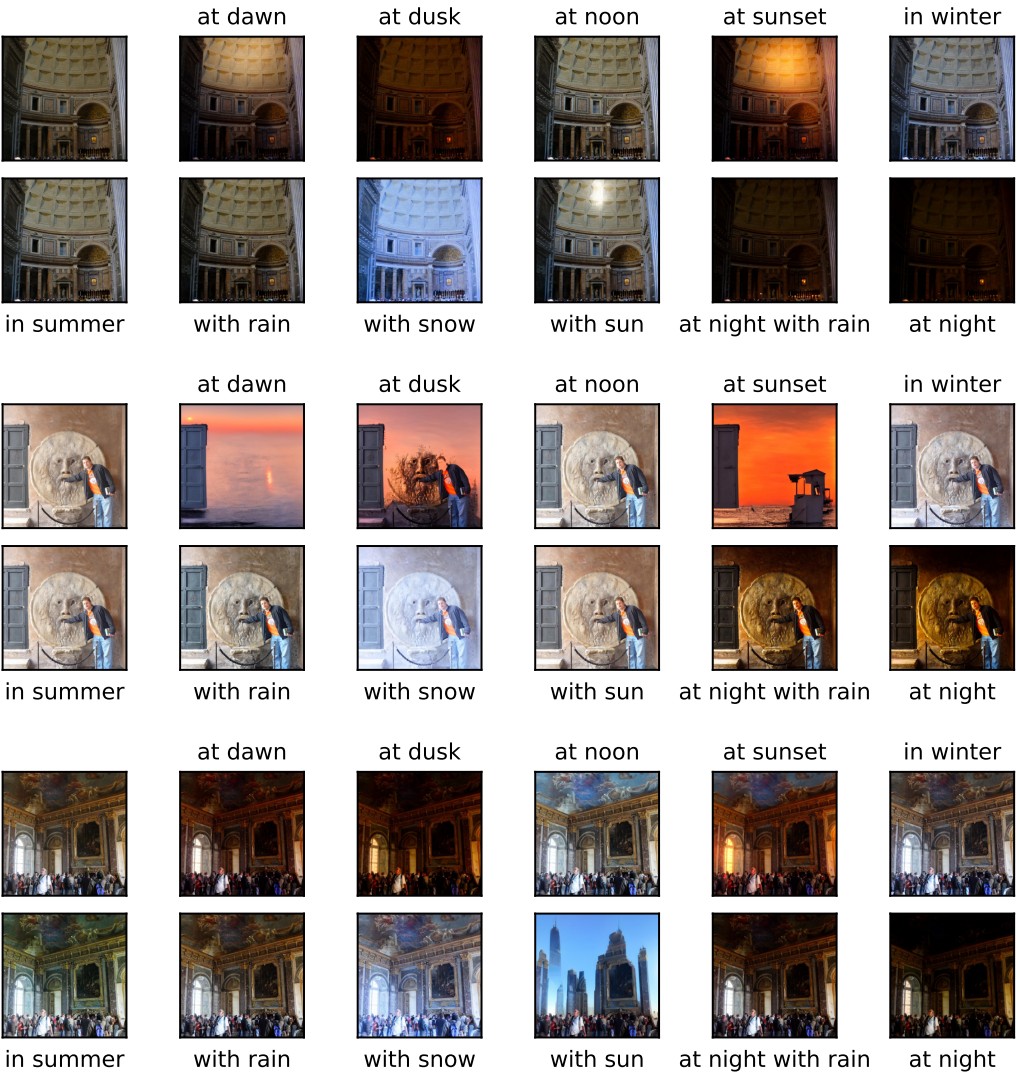

Figure 8: **Peculiar synthesis and failure cases** for training images not suited to the prompts.

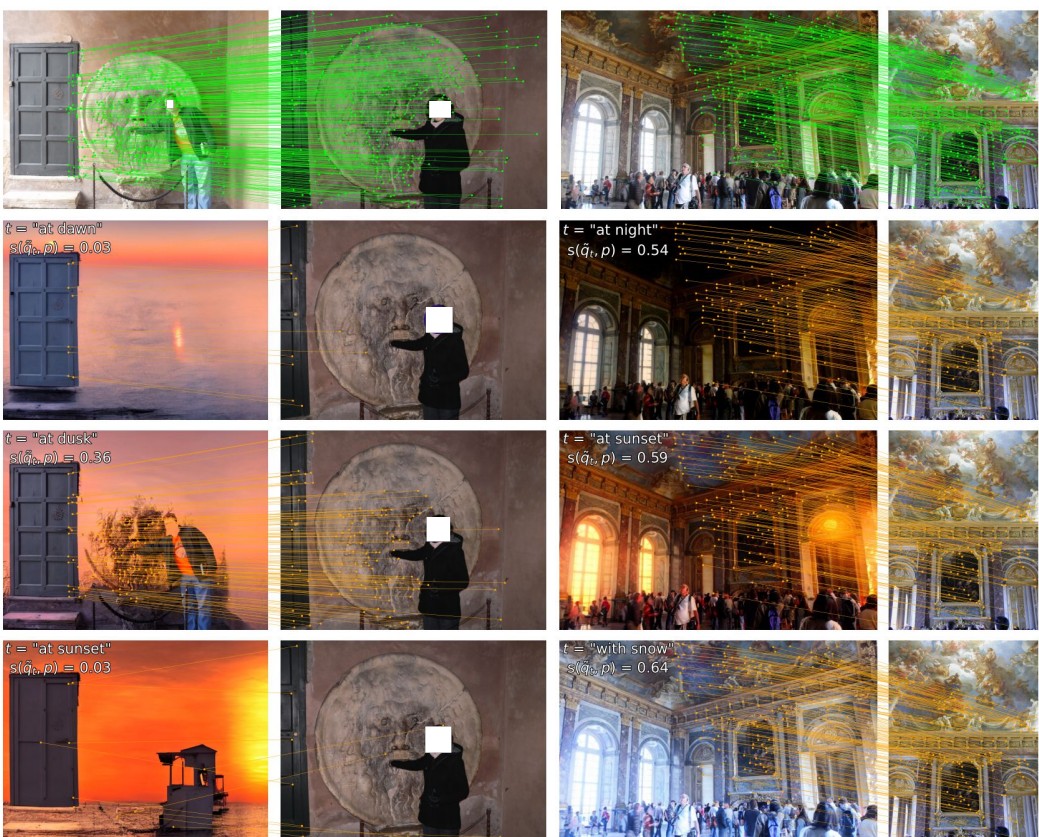

Figure 9: **Geometric verification for indoor image variants**. The corresponding prompt and score $s$ are printed on each pair. Matches are discovered with the LightGlue (Lindenberger et al., 2023) algorithm.

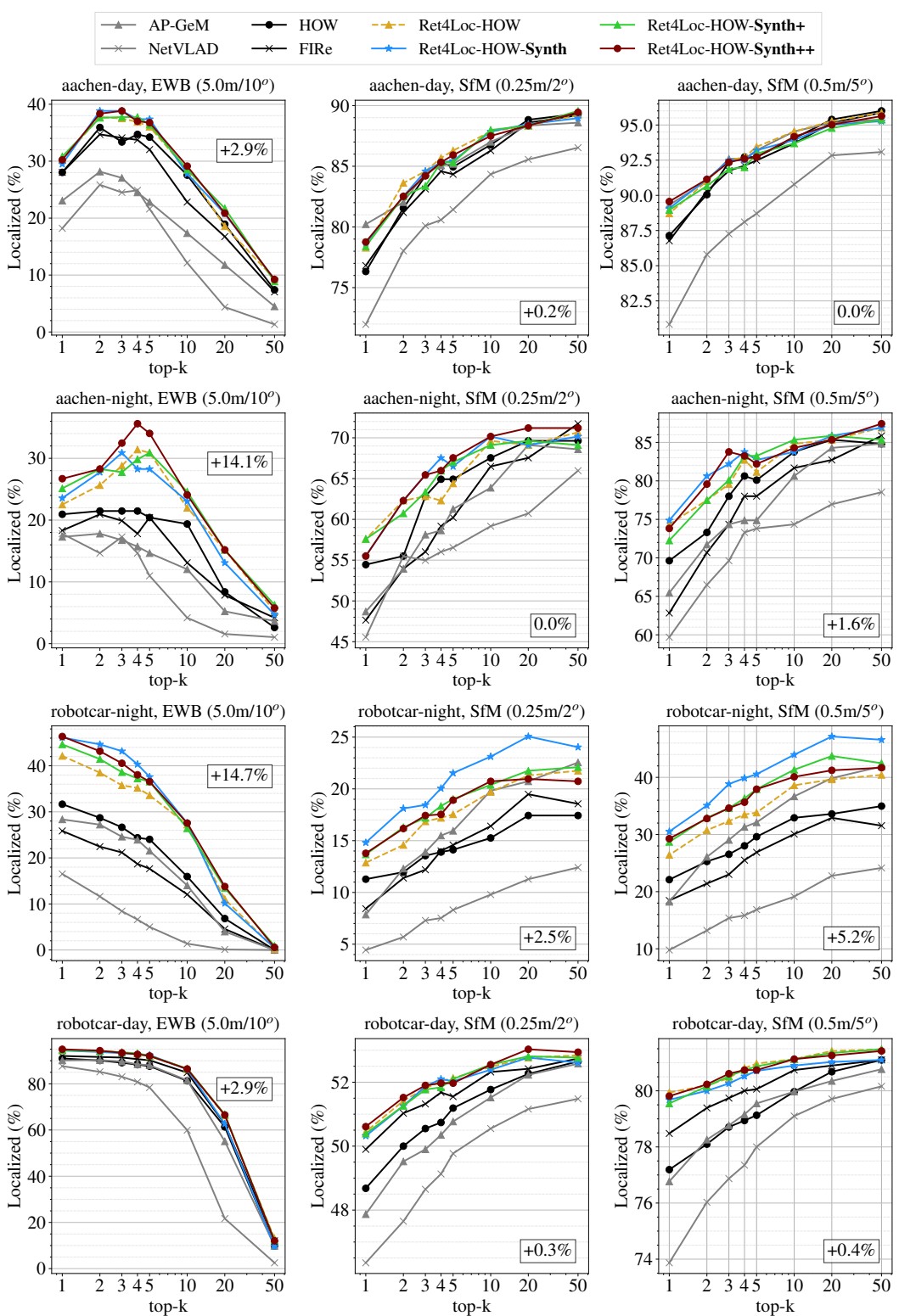

Figure 10: **Visual localization results on Aachen Day-Night v1.1 and Robotcar Seasons (v1)**.

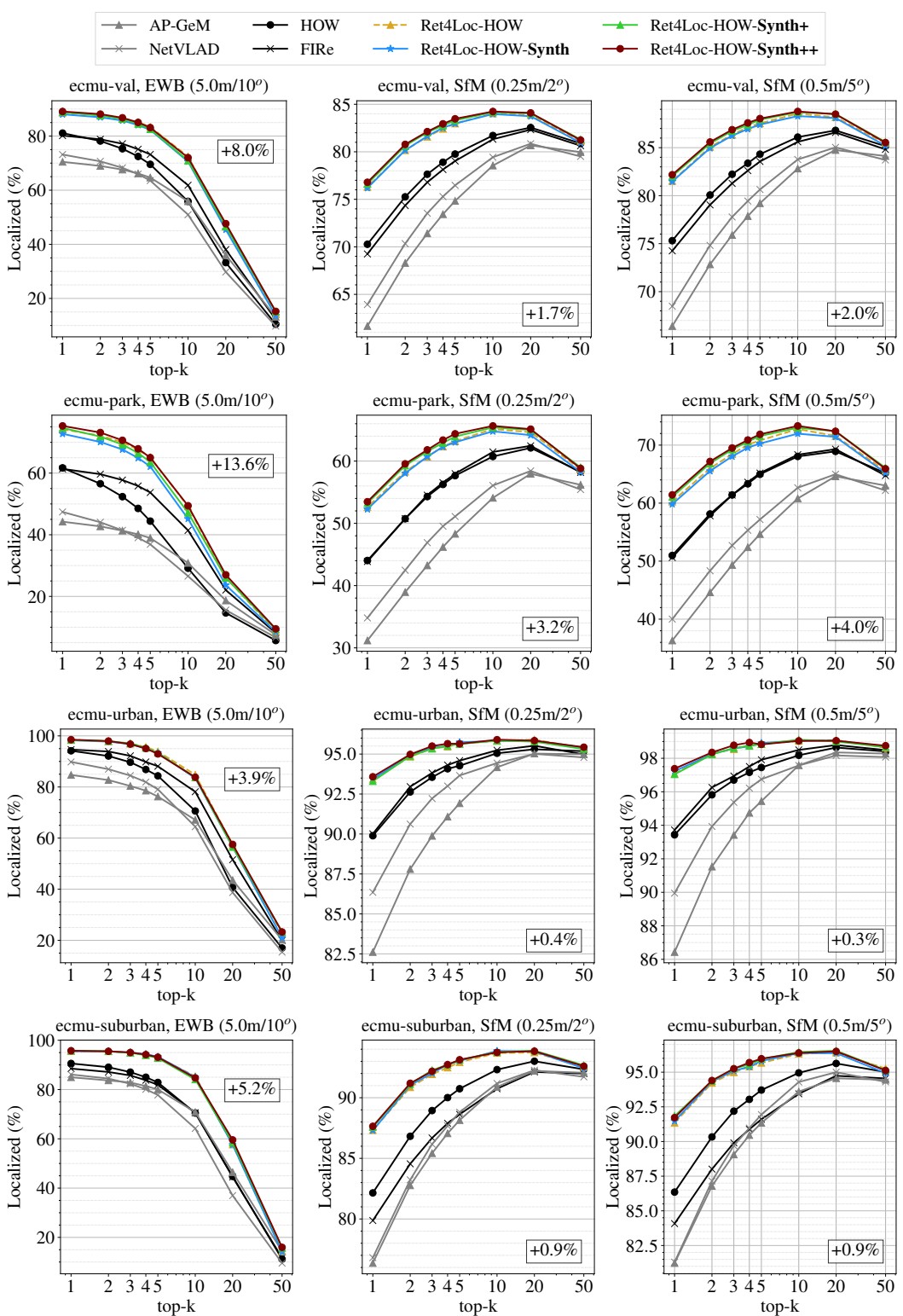

Figure 11: **Visual localization results on four splits of the Extended CMU Seasons dataset**.

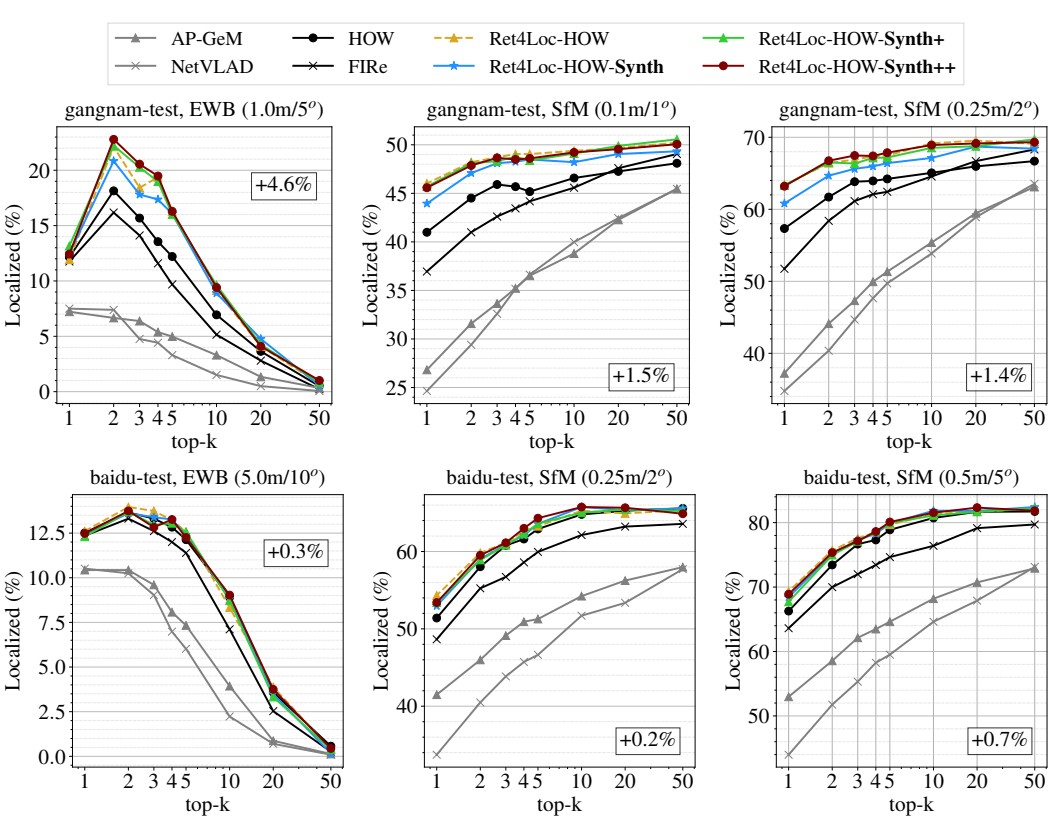

Figure 12: **Visual localization results on Baidu Mall and Gangnam Station B2 datasets**.

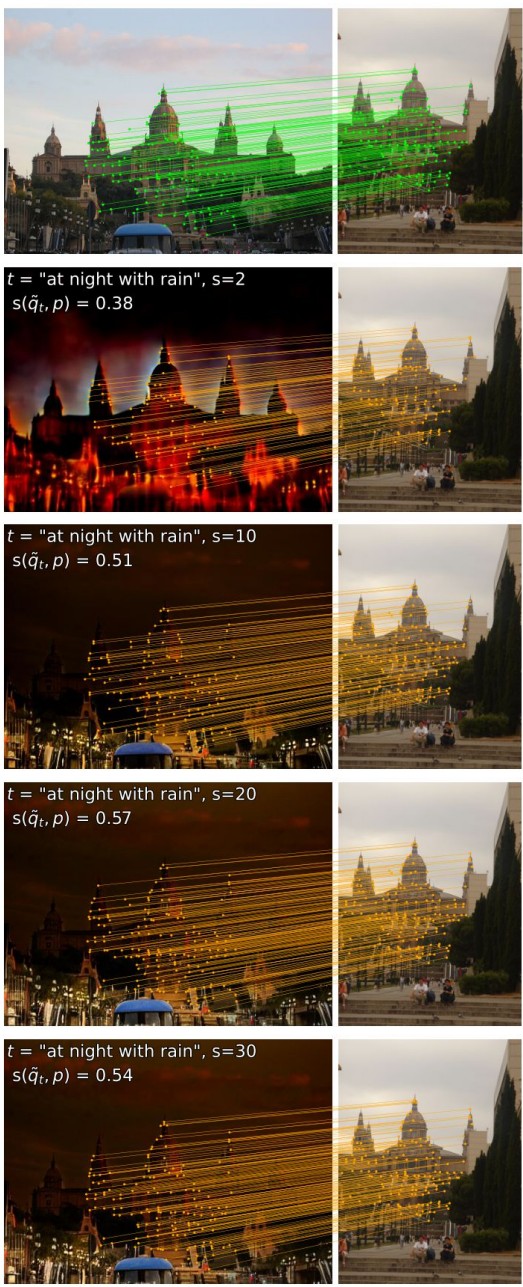

Figure 13: **Geometric matching for image generation after different steps.** We see that the number of matches is generally consistent for steps between 10 and 30. For this visualization we increase the number of features used by LightGlue to 2048.

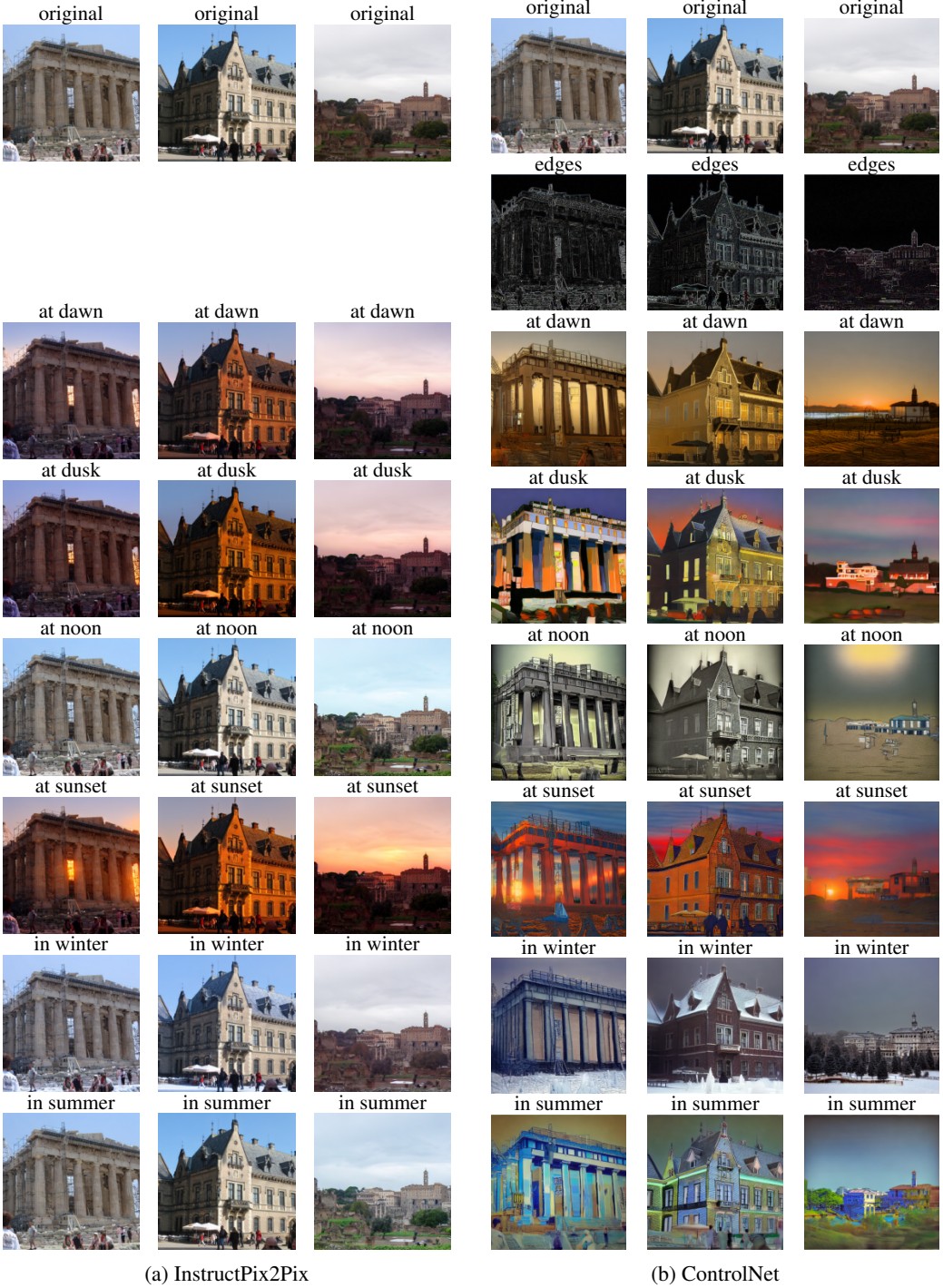

(a) InstructPix2Pix          (b) ControlNet

Figure 14: **Synthetic images from InstructPix2Pix (Brooks et al., 2023) (left) and Control-Net (Zhang et al., 2023) (right)**. For a given image ("original") and the 11 prompts we consider in this work, for ControlNet, we generate images using edge maps obtained from the original images. See Fig. 15 for the rest of the prompts.

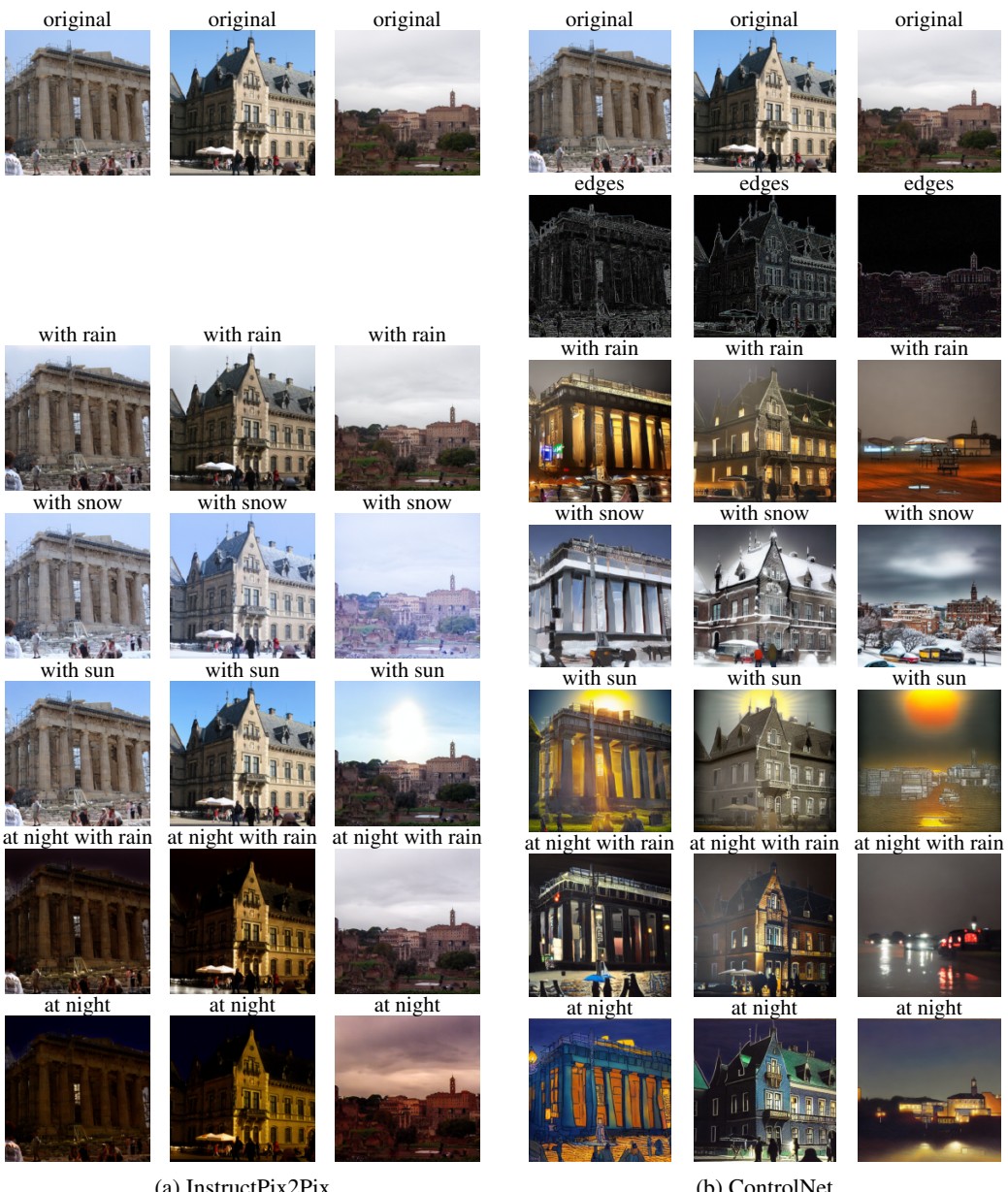

(a) InstructPix2Pix                    (b) ControlNet

Figure 15: Continuation of Fig. 14.

