# OpenReview forum: "Weatherproofing Retrieval for Localization with Generative AI and Geometric Consistency"
_ICLR.cc/2024/Conference — ICLR 2024 poster_

### Official Review · Reviewer_Hu6J · 2023-10-25

**Soundness:** 3 good
**Presentation:** 3 good
**Contribution:** 3 good
**Rating:** 8
**Confidence:** 3

**Summary:**

This paper tailors the training of a retrieval model to the challenging task of long-term visual localization. The paper uses language-based data augmentation to improve the robustness of retrieval models to adversarial daytime, weather, and seasonal variations, and further introduces geometry-based strategies for filtering and sampling from a pool of synthetic images. Experimental results show that the proposed method achieves significant improvements over the state-of-the-art on six common benchmarks.

**Strengths:**

1. It makes sense to me to start with state-of-the-art techniques for landmark retrieval and gradually adapt them to intuitions about visual localization. The paper uses language-based data augmentation to improve the robustness of retrieval models to adversarial daytime, weather, and seasonal changes.
2. The paper is well written and structured
3. The paper performs nice ablation experiments on all different combinations of pre-trained targets and dataset combinations.
4. The paper provides a detailed explanation and description of related work.

**Weaknesses:**

1. Relative to HOW (Tolias et al., 2020), InstructPix2Pix (Brooks et al., 2023), LightGlue (Lindenberger et al., 2023), the contributions appear to be incremental, such as two contributions training with synthetic variants, and synthetic variant generation and geometric consistency.
2. To handle large environments, a trade-off must be made between accuracy and computation time. Then, is the approach utilized in practical applications?

**Questions:**

1. What is the difference between the loss function of Ret4Loc-HOW-Synth (Eq. (2)) and Eq. (1)? Does Eq. (2) just use the extended tuple set?
2. How does the training time gets impacted with the newly introduced loss?
3. Since the authors discussed synthesizing a training set from generated text to image models, did they also compare on a visual classification task?

---

> ### Author Response · Authors · 2023-11-17
> **Author response**
>
> We sincerely thank the Reviewer for finding our approach intuitive, our paper well written and for praising our experimental evaluation. Below we respond to all their questions and concerns.
>
> > “To handle large environments, a trade-off must be made between accuracy and computation time. Then, is the approach utilized in practical applications?”
>
> Our method proposes a better training process designed to improve the model used in the retrieval step of visual localization. All the changes we make are at _training time_, and do not affect scalability at test time. In fact, it is worth noting that improving the retrieval step is what makes the difference in large environments, i.e. localization for cases where the database is large. The shortlist passed on to the pose approximation methods matters even more in such cases. We hope this answers the Reviewer’s question and apologize if we misunderstood what the reviewer meant by “large environments”. We are happy to discuss this further if needed.
>
> > “What is the difference between the loss function of Ret4Loc-HOW-Synth (Eq. (2)) and Eq. (1)? Does Eq. (2) just use the extended tuple set?”
>
>
> There are two differences between the two losses: First, as the Reviewer notes, Eq. (2) extends the tuple set to all real and synthetic tuples. Apart from that, it also introduces the feature aggregation function phi, a function that aggregates the embeddings from the real and all synthetic variants per image. Because of that, tuples are not treated independently anymore at the loss level.
>
> > “How does the training time gets impacted with the newly introduced loss?”
>
>
> This is a great question. Below we report training time and memory requirements for our method and the baseline. It is worth highlighting that we can train our model in less than 8 hours on a single A100 GPU. We ran image generation as a pre-processing step once, this took about 25 hours for the SfM-120k dataset when running it for all prompts in parallel.
>
>
> | Model                | Number of variants K used in Eq. (2) | Total number of available training images |  Training time | Maximum GPU memory usage | RobotCar Night Recall@1 | RobotCar Night Recall@10 |
> | ---------------------------------- | --------------------------------- | ----------------------------------------- |  ------------- | ------------------------ | ----------------------- | ------------------------ |
> | Ret4Loc-HOW         | 1 (no synthetic)                     | 91K (real)                                       | ~4 hours      | 21.8 GB                  | 24.1                    | 38.1                     |
> | Ret4Loc-HOW-Synth++ | 3                                    | 91K (real) + 1M (synthetic)                                      | ~8 hours      | 65.4 GB                  | __33.3__ (+9.2%)            | __45.3__ (+7.2%)             |
>
>
> > Since the authors discussed synthesizing a training set from generated text to image models, did they also compare on a visual classification task?
>
>
> Part of the motivation of our paper is to tailor retrieval to the specific task of visual localisation and all our experiments and backbones are tuned for that. We believe it is beyond the scope of this paper to also report results on classification. However, there do exist recent papers that use a training set from generated text to image models for ImageNet classification (e.g. Sariyildiz et al, CVPR 2023 or He et al, ICLR 2023) that we could discuss further if the reviewer believes we should.
>
> We thank the Reviewer again for a constructive review and hope that our answers above clarify all main concerns. We are looking forward to hearing their thoughts on our responses and welcome further discussion.

---

### Official Review · Reviewer_dtiA · 2023-10-30

**Soundness:** 2 fair
**Presentation:** 3 good
**Contribution:** 2 fair
**Rating:** 6
**Confidence:** 5

**Summary:**

This paper aims to improve the retrieval accuracy to enhance the localization performance under challenging conditions such as illumination, weather and season variations between the query and reference images.

To deal with these challenges, the authors propose to expand the training data by generating a large number of synthetic images with the aforementioned variations via generative text-to-image models. Additionally, a geometry-based method is adopted to filter and sample training images.

The model trained on expanded training data gives better performance than previous methods on multiple public datasets.

**Strengths:**

1.	Motivation. A map used for localization is usually built on images captured at very short the time interval without large appearance changes. However, the query images could be recorded at different times with huge illumination, weather, and season differences to reference images in the map, which makes long-term localization challenging.
In the paper, the authors propose to bridge the gap by generating training data with these variations. The idea is straightforward but very useful especially for the retrieval task.

2.	Geometric constraints. I am more interested in this part. One reason is as mentioned in the paper that this allows synthetic images to preserve location-specific information which cannot be guaranteed by generative models. One more important reason is that localization using 3D map has two steps: coarse localization by retrieval and fine localization with PnP + RANSAC from 2D-3D matches. Due to appearance changes, the best retrieved images may not give the most accurate poses as the final pose is also influenced by how matches can be found between the query and retrieved images. It seems like this point is not discussed in the paper.

3.	Extensive experiments. The proposed method is evaluated on multiple public datasets including RobotCar-Seasons, Aachen Day-Night, ECMU, Gangnam Station B2, Baidu Mall and Tokyo 24/7. Results demonstrate that regular augmentations are the most effective techniques (Ret4Loc-How vs. How) and the proposed training with synthetic data (Ret4Loc-How-synthetic) and geometric constraints (Ret4Loc-How-synth+ and Ret4Loc-How-synth++) also improve the performance.

**Weaknesses:**

The proposed method is simple and easy to follow. My major concerns come from the evaluation.

1.	Results on RobotCar-Seasons dataset. The textural prompts are from tags in RobotCar-Seasons dataset, but in Table 1, models + synthetic data + geometric constraints, denoted as Ret4Loc-How-synth+ and Ret4LocHow-synth++ do not give obvious improvements to Ret4Loc-How.

2.	In Table 2, Ret4Loc-HOW-Synth++ achieves better results than previous single-state methods but gives worse accuracy than approaches with reranking. Since in the paper the authors mention that reranking can also be applied to the proposed methods, why not provide results of proposed methods with reranking? In theory, models such as Ret4Loc-HOW-Synth++ plus reranking should work better than other methods with re-ranking.

3.	Comparison with NetVLAD and GEM. As far as I know, NetVLAD and GEM are the most popular methods used for retrieval on ECMU, Aachen and RobotCar-Seasons datasets. It would be better to include results of these methods. By the way, although ECMU contains images with diverse seasonal changes, it provides sequential images, which makes retrieval and localization easy. Therefore, results on Aachen and RobotCar-Seasons datasets are more convincing.

4.	Results of localization with 3D map. Fig.10 in appendix shows results with 3D map. However, according to the website (https://www.visuallocalization.net/benchmark/), some more recent methods such as SP+SG, sfd2_4000kpts_netvlad50, KAPTURE-R2D2-APGeM, imp_4kpoints_netvlad50, SP+LightGlue give more much better accuracy with only NetVLAD for reference search. As these methods give the current SOTA results, the proposed method in the paper should be compared with these methods. I can understand that these methods may use different local features and global features to R2D2 and HoW. But it is possible to replace the global feature with Ret4Loc-synth++ and report the results.

5.	It is not very clear to me in Fig. 9, the authors show matches given by LightGlue not R2D2, but R2D2 is the local feature used for finding 2D-2D matches in the paper. I also want to know is R2D2 used for geometric constraints?

6.	In Sec.5, related works of visual localization under challenging conditions are discussed. Image augmentation with style transfer such Day-Night transfer is only one way to handle appearance changes. Some other solutions like semantic localization and semantic features (e.g., Visual Localization Using Sparse Semantic 3D Map, Sfd2). Related papers can be found from https://www.visuallocalization.net/benchmark/) are also solutions to challenging conditions as semantics are more robust to appearance changes. Discussions on these works should also be included if this work focuses on improve visual localization under challenging conditions.

**Questions:**

Please see Weaknesses for details.

---

> ### Author Response · Authors · 2023-11-17
> **Author response - part 1 of 2**
>
> We thank the Reviewer for lots of constructive feedback and for finding strengths in our work across multiple dimensions. In this response we address the concerns that the reviewer raised regarding our evaluation.
>
> > 1. “Results on RobotCar-Seasons dataset. The textural prompts are from tags in RobotCar-Seasons dataset, but in Table 1 [...] geometric constraints do not give obvious improvements over Ret4Loc-HOW.”
>
> It is true that, for the day split of RobotCar, we do not see gains from the use of synthetic data. This is however not the case for the more challenging Robotcar night split. There, the best model is Ret4Loc-HOW-Synth, i.e. the one that uses all synthetic data without filtering (row 3). It shows strong performance gains over Ret4Loc-HOW, i.e. 3.3% and 6.7% in localization accuracy for two precision thresholds using 3D maps, while gains are even higher for the EWB protocol, where retrieval matters the most.
>
> It is however also true that adding geometric constraints to filter/sample the data does not really improve over using all synthetic data for Robotcar in general. One explanation could be indeed the fact that textural prompts are coming from tags in RobotCar-Seasons: even distorted night images might act as regularization and lead to improved performance.
>
> However, this behavior is the exception: We can see from the extended result over all localization and retrieval datasets (Tables 1,2, and Figures 10,11,12) that top performance in most cases comes from our variants that also exploit geometric constraints.
>
> > 2. “In Table 2, Ret4Loc-HOW-Synth++ achieves better results than previous single-state methods but gives worse accuracy than approaches with reranking. [...] why not provide results of proposed methods with reranking? “
>
> We thank the Reviewer; we agree that having results with reranking in Table 2 would strengthen our paper even more. It will be hard to get those results by the end of the rebuttal period, but we will set up and run reranking experiments, and present them in a camera-ready version.
>
> That said, we believe that reranking experiments would be good to have, but not crucial. One of our paper’s contributions is our extensive evaluations on 5 localization datasets using precise visual localization with 3D maps (Table 1 and Figure 3 in the main paper, as well as Figures 10, 11 and 12). This is unique for a paper proposing a retrieval method yet essential in our case since part of our novelty is tailoring retrieval to localization. These localization evaluations that we report already go way beyond reranking experiments, which could be seen as an intermediate set evaluation.
>
>
>
> > 3. “Comparison with NetVLAD and GEM.”
>
> This is a great point and we apologize for these missing comparisons to standard methods. We have already computed the two methods for EWB (that is faster) and promise to report them in Fig 3 of the updated pdf we post during rebuttal. For EWB, we observe that the missing baselines change our relative gains reported in Fig 3 only slightly, and only for the Aachen-night EWB case (+14.1% -> +13.6%). In all other cases our strong gains remain unchanged, i.e. for Robotcar Night (+14.7%) and Gangnam (+4.6%). We also have results for Aachen-night SfM where our relative gains also remain unchanged (+1.6%). We will provide complete comparisons to both methods on all datasets for a camera-ready version.

---

> > ### Author Response · Authors · 2023-11-17
> > **Author response - part 2 of 2**
> >
> > > 4. “Results of localization with 3D map. [...] according to the website (https://www.visuallocalization.net/benchmark/), some more recent methods such as SP+SG, sfd2_4000kpts_netvlad50, KAPTURE-R2D2-APGeM, imp_4kpoints_netvlad50, SP+LightGlue give more much better accuracy with only NetVLAD for reference search. As these methods give the current SOTA results, the proposed method in the paper should be compared with these methods. “
> >
> > The Reviewer makes valid points regarding the results reported in the leaderboards of the visual localization benchmark. We have extensively looked into this during the past days, and after reproducing KAPTURE-R2D2-APGeM in our Kapture setup, i.e. with the exact same pose approximation and 3D COLMAP that we used for all methods in our experiments, we found  inconsistencies in the evaluation protocols in that benchmark with respect to retrieval. For example, we use the whole Robotcar dataset as the database; instead, that specific leaderboard conducts retrieval in each split separately. The latter makes the retrieval part _much easier_ and less important. We instead follow the protocol used in Humenberger et al (IJCV 2022) which is a recent review on retrieval on localization. The numbers we get for ApGeM and NETVLAD match that paper’s reported numbers. Our results are therefore not directly comparable to those presented on the site’s leaderboard.
> >
> > Our paper focuses on studying and improving the retrieval part of visual localization under a fair setting, i.e. we consider the pose estimation step fixed and common to all results. This is why we used a basic method like R2D2, and we only vary the model used in the retrieval step. In that regard, our paper is unique among papers that propose a new retrieval method: it is the first retrieval paper that goes the extra mile and extensively evaluates localization on 5 common benchmarks for different top-k, and for both for the EWB and 3D map protocols. This is unlike the closest related works that only measure retrieval on place recognition datasets and in the best case report top-1 EWB performance for localization.
> >
> > We hope that the Reviewer agrees that our paper’s contributions and novelty are strong, even without getting results that rival the top entries in this leaderboard. This leaderboard favors methods that perform really precise pose approximation, a step independent from retrieval, which is hence out of the scope of our research question. We are more than happy to discuss this further if needed.
> >
> > > “5. It is not very clear to me in Fig. 9, the authors show matches given by LightGlue not R2D2, but R2D2 is the local feature used for finding 2D-2D matches in the paper. I also want to know is R2D2 used for geometric constraints?”
> >
> > We only use LightGlue for validating real-synthetic image matching pairs that are used for training the retrieval model.This means that it is only used for verification purposes during the construction of the extended training set. No synthetic images are used during localization, so LightGlue is not at all used at test time. Since we needed to validate a large number of pairs, we found LightGlue to be both fast and highly robust. We never tested R2D2 for this task.
> >
> > We apologize for any confusion, we will clarify this in the text.
> >
> >
> >
> > >”6. In Sec.5, related works [proposing] solutions like semantic localization and semantic features (e.g., Visual Localization Using Sparse Semantic 3D Map, Sfd2). [...] are also solutions to challenging conditions as semantics are more robust to appearance changes. Discussions on these works should also be included if this work focuses on improving visual localization under challenging conditions.”
> >
> > This is a good point, thank you for the suggestion.
> > We will add and discuss them in the related work section.
> >
> > We thank the Reviewer again for a constructive review and hope that our answers above clarify all main concerns. We are looking forward to hearing their thoughts on our responses and welcome further discussion.

---

> > > ### Comment · Reviewer_dtiA · 2023-11-21
> > > **Reply to Q4 and Q5**
> > >
> > > Q4
> > >
> > > I agree that the idea in the paper is simple and useful. Because of this, I hope to see a more convincing comparison with previous SOTA methods that achieve the best localization performance on Aachen and RobotCar datasets.
> > >
> > > Previous methods do the retrieval on the whole dataset of Aachen, so it would be a fair comparison to replace NetVLAD with the method proposed in the paper and report the numbers.
> > >
> > > As for RobotCar dataset, the author could adopt the same setup used by prior approaches and conduct the experiments.
> > >
> > > Q5.
> > >
> > > Thanks for the reply. One question which is not answered is that if R2D2 or LightGlue+other local feature is used in the training process especially the geometric constraints.

---

> > ### Comment · Reviewer_dtiA · 2023-11-21
> > **3. Comparison with NetVLAD and GEM**
> >
> > Many thanks for the reply. It is very easy to find the source code of NetVLAD and GeM from github to run the experiments. The author can also use the open-sourced framework HFNet which already incorporates NetVLAD and GeM to do that. I believe it will not take much time.

---

> ### Author Response · Authors · 2023-11-21
> **Additional authors' response**
>
> >Comparissons to AP-GeM and NetVLAD added for Aachen and Robotcar
>
> We have added Ap-GeM and NetVLAD in Figures 3 and the extended Fig 10 for Aachen and Robotcar. We see that Ret4Loc models still offer significant gains over all baselines on the night slices of the two important benchmarks that the reviewer mentions. Performance gains are noteworthy in the EWB case, i.e. the case where retrieval plays a primary role. We report gains of _over 10% in localization accuracy_ with our top Ret4Loc model for both Aachen night and Robotcar night.
>
> Let us also note that we have verified our reproductions match the results reported for the two methods in the IJCV 2022 study on retrieval for localization by Humenberger et al. (see Fig 6 of the journal paper).
>
>
> > Q4. I agree that the idea in the paper is simple and useful. Because of this, I hope to see a more convincing comparison with previous SOTA methods that achieve the best localization performance on Aachen and RobotCar datasets. Previous methods do the retrieval on the whole dataset of Aachen, so it would be a fair comparison to replace NetVLAD with the method proposed in the paper and report the numbers.
>
> We thank the reviewer again for finding the idea simple and useful. We hope that after adding the two baselines the reviewer is also convinced that our method also offers strong gains in terms of both retrieval  and localization performance. Let us note again that all our localization experiments are run _under a fair setting_, i.e. exactly the setup that the reviewer describes: We are only switching the retrieval method and keep the pose estimation step fixed in all experiments to a common choice for SfM based- localization, i.e. Kapture with Feather-R2D2.
>
> > As for RobotCar dataset, the author could adopt the same setup used by prior approaches and conduct the experiments.
>
> Despite the fact that in that setup retrieval would matter even less, we agree that this is another experiment that would be nice to have for completeness. We hope however that the reviewer understands that all these experiments are hard to conduct during rebuttal. Similar to the experiment that the reviewer asked with reranking, we do intend to include it in a camera ready version if our paper gets accepted.
>
> We also hope the reviewer agrees that either of the two are merely complementary to the strong and fair evaluations that already exist in the paper.
>
> > Q5. Thanks for the reply. One question which is not answered is that if R2D2 or LightGlue+other local feature is used in the training process, especially the geometric constraints.
>
> We apologize if our response above was not clearly expressed. We use only LightGlue during training, for computing the geometric consistency score for each of the synthetic pairs in our extended _training set_ (i.e. SfM-120k). We then use R2D2 local features only in the pose estimation step of localization via 3D maps, i.e. as part of the Kapture based SfM-based localization process that we consider a black box. We chose R2D2 since this is a common choice that is also used in the recent study on retrieval for localization by Humenberger et al (IJCV 2022).
>
> Thank you for the interaction and discussion, we appreciate it. Let us know if the above additional response clarifies all remaining concerns. Happy to discuss further.

---

> > ### Comment · Reviewer_dtiA · 2023-11-22
> > **pose rebuttal**
> >
> > Thanks for the reply. I have no other concers.

---

### Official Review · Reviewer_oYk5 · 2023-10-30

**Soundness:** 3 good
**Presentation:** 3 good
**Contribution:** 3 good
**Rating:** 6
**Confidence:** 3

**Summary:**

In this paper, the author proposes a new method named for Long-term Visual localization. The author first builds a strong baseline, i.e., a baseline with strong data augmentations for visual localization. Then, the author proposed to use the conditional diffusion model to generate synthetic data, then the author proposed some strategies to train with joint real data and synthetic data. The proposed methods achieve performance gain over previous methods.

**Strengths:**

In general, I'm satisfied with the paper. The proposed method is simple and effective. Without bells and whistles, the proposed method achieves state-of-the-art performances on major benchmark datasets.

**Weaknesses:**

However, I still have some concerns about the paper:

1. Though the proposed method is simple, I think the author could provide more detailed comparisons of the method, like the number of training images, the generating time, etc. The proposed method utilizes a conditional diffusion model to generate the augmented dataset rather than the previously manual data augmentations. Thus it is noticeable and essential to bring the cost (training and memory cost), the training image numbers comparisons, and the training time cost for the model.

2. Though the InstructPix2Pix model is a natural choice for the paper. There are many choices with the model, e.g., Stable Diffusion with ControlNet, etc., the author could provide more extra experiments with different methods (even though ablations on a small dataset) will make the paper more robust. Also, the proposed filtering metric should be discussed over different generation settings.

**Questions:**

Please mainly see the weaknesses section for details.

---

> ### Author Response · Authors · 2023-11-17
> **Author response - part 1 of 2**
>
> We thank the Reviewer for a constructive and overall positive review. In this response, we address the reviewer’s concerns and provide the data and statistics requested. We will incorporate all information presented below in the updated pdf.
>
> > 1.  “more detailed comparisons of the method, like the number of training images, the generating time [...] it is noticeable and essential to bring the cost (training and memory cost), the training image numbers comparisons, and the training time cost for the model.”
>
> Thank you for this comment, we fully agree that it is important to report such statistics.
> We present and discuss i) the model training and memory usage, and ii) the generation part separately, as, in practice, we ran synthetic image generation _as an offline process_ once and saved all 11 variants for each training image.
>
> The first table below presents timings for model training and memory usage, together with some basic quantitative results for the retrieval performance on Robotcar. Training times in this table  _do not include the part of the image generation itself_ and were obtained using a single A100 GPU.
>
> | Model                | Number of variants K used in Eq. (2) | Total number of available training images |  Training time | Maximum GPU memory usage | RobotCar Night Recall@1 | RobotCar Night Recall@10 |
> | ---------------------------------- | --------------------------------- | ----------------------------------------- |  ------------- | ------------------------ | ----------------------- | ------------------------ |
> | Ret4Loc-HOW         | 1 (no synthetic)                     | 91K (real)                                       | ~4 hours      | 21.8 GB                  | 24.1                    | 38.1                     |
> | Ret4Loc-HOW-Synth++ | 3                                    | 91K (real) + 1M (synthetic)                                      | ~8 hours      | 65.4 GB                  | __33.3__ (+9.2%)            | __45.3__ (+7.2%)             |
>
> The second table below presents the synthetic image generation time. It is affected by two parameters: the generated image resolution (Resolution) and the number of diffusion steps (Steps). Below we show the generation time per image in seconds (Time), for different resolutions and number of steps.
>
> | Resolution | Steps | Time (sec) |
> | ---------- | ------------------- | ----------------------------- |
> | 512        | 10                  | 0.30                          |
> | 512        | 20                  | 0.44                          |
> | 512        | 30                  | 0.64                          |
> | 768        | 10                  | 0.62                          |
> | _768_        | _20_                  | _0.98_                          |
> | 768        | 30                  | 1.39                          |
> | 1024       | 10                  | 1.16                          |
> | 1024       | 20                  | 1.99                          |
> | 1024       | 30                  | 2.78                          |
>
> We chose to use a resolution of 768 pixels and 20 steps for all results in the paper. For these parameters, the time needed for generating all synthetic images for a single prompt was approximately 25 hours for the complete SfM dataset. Given that the process is independent per prompt, we ran synthetic data extraction in parallel for all prompts and generated images for all variants _in about a day_.  With model training taking approximately 8 hours, we see that data generation time is at least 3 times higher than the actual model training. However, we do not believe this is an issue in practical applications for mainly three reasons:
>
> a) As a pre-processing step, data generation needs to run only once. This is unlike model training, a process that one usually needs to run multiple times, e.g. to perform hyperparameter validation.
>
> b) the overall training time for our model (data generation + model training) is approximately 32 hours. This means that learning state-of-the-art retrieval models for visual localization using our method is manageable in reasonable time, even with modest resources.
>
> c) We believe that over time the data synthesis overhead will eventually be even less of an issue: generative AI models will inevitably become more efficient and we envision in the near future such generation to happen on-the-fly, during batch construction.

---

> > ### Author Response · Authors · 2023-11-17
> > **Author response - part 2 of 2**
> >
> > > 2. “Though the InstructPix2Pix model is a natural choice for the paper. There are many choices with the model, e.g., Stable Diffusion with ControlNet”
> >
> > Thank you for another valid comment. While experimenting with InstructPix2Pix, we also tested ControlNet. We found the latter to produce images far more stylized and we deemed them unfit for our purpose. We uploaded synthetic images generated via iP2P and ControlNet at the following anonymous link, for the images depicted in Fig 2:
> >
> > https://imgur.com/a/LCBsDxr
> >
> > We will add these qualitative results in the appendix of our updated pdf.
> >
> > > Also, the proposed filtering metric should be discussed over different generation settings.
> >
> > This is also a great comment, thank you. The most important generation parameter that controls quality (and extraction time) is the number of steps. We intend to include qualitative results for geometric verification with synthetic images generated with different numbers of steps in the updated pdf.
> >
> > We thank the Reviewer again for a constructive review and hope that our answers clarify all main concerns. We are looking forward to hearing their thoughts on our responses and welcome further discussion.

---

> > > ### Comment · Reviewer_oYk5 · 2023-11-22
> > > **Reply to authors**
> > >
> > > I think the response has adequately addressed my concerns in the review. Thanks the author for providing detailed experiments and explanations.

---

### Author Response · Authors · 2023-11-17
**Authors general response to reviews**

We thank the Reviewers for their constructive feedback.

We are happy that all Reviewers praised our intuitive and effective method (Reviewer oYk5: “simple and effective”, Reviewer dtiA: “straightforward but very useful”, Reviewer Hu6J: “makes sense to me”), the fact that we are reporting state-of-the art results on common benchmarks (Reviewer oYk5, Reviewer dtiA), and our extensive validation (Reviewer dtiA: “extensive experiments”, Reviewer Hu6J: “nice ablations”). We are also glad that reviewers found the paper “well written and structured” (Reviewer Hu6J) and with “detailed explanation and description of related work” (Reviewer Hu6J).

In individual answers below, we respond to all comments and questions from each Reviewer. We hope that posting our responses early will encourage further discussions and interaction with the Reviewers. We will upload an updated pdf with all requested changes before the end of the discussion period.

We hope that our responses clarify all important concerns and we are looking forward to feedback and additional questions if some remain.

---

### Author Response · Authors · 2023-11-21
**Updated pdf with revision after rebuttal was uploaded**

We uploaded an updated pdf with the following changes/additions (colored in blue to make it easier to follow):
- Added training and generation times in the main paper
- Added NetVLAD and AP-GeM as baselines for most dataset in Fig 3 as well as the extended results in Fig 10-11-12.
- Added appendix B5 discussing training and generation times in detail.
- Added some qualitative results of geometric matching using different diffusion steps during  InstructPix2Pix generation (Fig 13)
- Added qualitative comparison of InstructPix2Pix generations to ControlNet (Fig 14 &15)
- Added a brief discussion about related works leveraging semantics to learn features robust to weather and seasonal conditions in the main paper, and a more detailed paragraph in Appendix D.

 All new text is clearly marked with blue color.

We want to thank all reviewers again for constructive feedback that made our submission even stronger. We hope that these changes, together with our detailed responses, help clarify all concerns the reviewers had. We are looking forward to any additional comments and to discussing any remaining issues.

---

### Meta-Review · Area_Chair_LYvA · 2023-12-28

**Metareview:**

Summary:
The paper proposes a novel approach for improving long-term visual localization performance by addressing challenges such as illumination, weather, and seasonal variations between query and reference images. The authors build a strong baseline with robust data augmentations and introduce a conditional diffusion model to generate synthetic data. They employ strategies for joint training with both real and synthetic data, resulting in enhanced performance compared to previous methods. Additionally, the authors focus on retrieval accuracy by expanding training data with synthetic images generated using generative text-to-image models. A geometry-based method is incorporated to filter and sample training images, leading to superior performance on multiple public datasets. The tailored training approach for long-term visual localization includes language-based data augmentation to enhance robustness against adversarial variations, and experimental results showcase significant improvements over state-of-the-art methods on six common benchmarks.

Strengths:
The paper introduces a straightforward yet highly valuable method for long-term visual localization, achieving state-of-the-art performance on major benchmark datasets. Motivated by the clear significance of the challenge, the proposed approach proves particularly effective for retrieval tasks. By incorporating geometric constraints, the method ensures that synthetic images preserve location-specific information, addressing a limitation of generative models. Extensive evaluations on multiple public datasets demonstrate the method's promising results and robustness. Leveraging language-based data augmentation enhances retrieval model robustness against adversarial conditions. Notably, well-executed ablation experiments cover diverse combinations of pre-trained targets and datasets, reinforcing the paper's credibility and thoroughness. Overall, the paper's clear motivation, practical methodology, and comprehensive evaluation contribute to its strength in advancing long-term visual localization.

Weaknesses:
The paper lacks some detailed comparisons of the proposed method, such as the number of training images and the generating time, which would be necessary to provide a more comprehensive understanding of its performance. Additionally, there may be a need for more extensive experiments with various generative model methods to assess the robustness and versatility of the proposed approach. The paper falls short in providing sufficient details about some experiment settings and results, also including some baseline comparisons in the benchmarks. There are concerns raised about the potential trade-off between accuracy and computation time, particularly considerations for practical applications. Furthermore, compared to previous works, the contributions of the paper seem incremental, particularly in the aspects of training with synthetic variants and addressing synthetic variant generation and geometric consistency.

**Justification For Why Not Higher Score:**

The paper proposes a simple yet effective method to tackle a very specific question for long-term visual location task. The comprehensive experiments on multiple benchmarks demonstrate the effectiveness of the proposed method, but the technical contribution may be incremental to some extent and the applicability of the method may be kind of limited to the target task.

**Justification For Why Not Lower Score:**

The proposed method is demonstrated to be effective by the comprehensive evaluation on multiple benchmarks. Although there are some weakness and concerns as mentioned above and also raised by reviewers, the rebuttal from the author has addressed these questions and concerns in a good way.

---

### Decision · Program_Chairs · 2024-01-16

Accept (poster)